# Color, activity period, and eye structure in four lineages of ants: Pale, nocturnal species have evolved larger eyes and larger facets than their dark, diurnal congeners

**Robert A. Johnson**[ORCID]*, **Ronald L. Rutowski**

School of Life Sciences, Arizona State University, Tempe, Arizona, United States of America

* Robert.Johnson4@asu.edu

## Abstract

The eyes of insects display an incredible diversity of adaptations to enhance vision across the gamut of light levels that they experience. One commonly studied contrast is the difference in eye structure between nocturnal and diurnal species, with nocturnal species typically having features that enhance eye sensitivity such as larger eyes, larger eye facets, and larger ocelli. In this study, we compared eye structure between workers of closely related nocturnal and diurnal above ground foraging ant species (Hymenoptera: Formicidae) in four genera (*Myrmecocystus*, *Aphaenogaster*, *Temnothorax*, *Veromessor*). In all four genera, nocturnal species tend to have little cuticular pigment (pale), while diurnal species are heavily pigmented (dark), hence we could use cuticle coloration as a surrogate for activity pattern. Across three genera (*Myrmecocystus*, *Aphaenogaster*, *Temnothorax*), pale species, as expected for nocturnally active animals, had larger eyes, larger facet diameters, and larger visual spans compared to their dark, more day active congeners. This same pattern occurred for one pale species of *Veromessor*, but not the other. There were no consistent differences between nocturnal and diurnal species in interommatidial angles and eye parameters both within and among genera. Hence, the evolution of eye features that enhance sensitivity in low light levels do not appear to have consistent correlated effects on features related to visual acuity. A survey across several additional ant genera found numerous other pale species with enlarged eyes, suggesting these traits evolved multiple times within and across genera. We also compared the size of the anterior ocellus in workers of pale versus dark species of *Myrmecocystus*. In species with larger workers, the anterior ocellus was smaller in pale than in dark species, but this difference mostly disappeared for species with smaller workers. Presence of the anterior ocellus also was size-dependent in the two largest pale species.

## Introduction

Ectotherms display extensive variation in color that arises at least in part from variation in the amount of pigment deposited in the cuticle/integument, with melanin being the most

---

**Data Availability Statement:** All data are deposited in the Dryad Digital Repository at doi:10.5061/dryad.nzs7h44v8.

**Funding:** The authors received no specific funding for this work.

**Competing interests:** The authors have declared that no competing interests exist.

common pigment [1, 2]. Diverse selective factors favor the evolution of dark body coloration including biotic factors such as predation, sexual selection, and disease resistance [3–8], and abiotic factors such as temperature, ultraviolet–B radiation, and desiccation [9–13]. Despite the various potential benefits of melanin deposition, numerous clades contain species with little or no pigment in their integument. These pale species occur in various taxa including fish, salamanders, insects, shrimp, and spiders [14–17].

One common correlate of pigment level in the integument is environment light level. Pale animals with little to no melanin are commonly active in environments with no ambient light such as caves, the deep sea, soil, and parasites inside the body of hosts [16, 18, 19], but are rare in terrestrial environments. Animals that live in dim light conditions, i.e., active nocturnally, sometimes also have little pigmentation and thus are pale compared to their diurnal congeners [e.g., bees, 20, 21]. However, the adaptive advantage of low pigment levels in low-light environments is unclear given that melanin is relatively cheap to produce and its potential advantages are many [22, 23, but see 24].

Species that deposit little pigment in their integument often display a suite of associated and selectively advantageous traits to low-light environments. One common adaptation in these species, especially those in lightless environments, is a severe reduction in or loss of eyes, with this trait being particularly well-studied in fish and other species that have pigmented terrestrial counterparts with fully developed eyes [14, 15, 25, 26]. Alternatively, many organisms that live in dim light environments and have lost some to most of their pigment possess exceptionally large eyes that enhance visual system performance in low light [20, 21].

Herein, we explore the association between body coloration, daily activity patterns, and eye structure in ant species that vary in the extent of their cuticular pigmentation. We designate two categories of coloration: pale (little pigmentation, appearing mostly whitish-yellow to yellowish to amber), and dark (extensive pigmentation, appearing orange or light to dark brown or black) (see Figs 1–4). Existing knowledge about the relationship between body color, light environment, and eye structure in ants suggests that they display relationships common in other taxa, i.e., (1) compared to close relatives from well-lit environments, species that live in lightless subterranean habitats are paler in color and have eyes that are absent or severely reduced in size [e.g., 27, 28], and (2) pale, nocturnal species that forage above ground have relatively large eyes compared to diurnally foraging species [e.g., 29]. Specifically, this study was motivated by the observation that eyes and facet lenses were larger in pale, nocturnal, above ground foraging species of honey pot ants (*Myrmecocystus* subgenus *Myrmecocytus*–subfamily Formicinae) compared to their dark congeners [see 30].

Broadly, we were interested in the evolution and consequences of these associations, and we used a comparative approach to examine these relationships in four ant genera in two subfamilies (*Myrmecocystus*—subfamily Formicinae and *Aphaenogaster*, *Temnothorax*, *Veromessor*—subfamily Myrmicinae) that contain pale and dark species. This multitaxa approach strengthened our ability to make evolutionary inferences.

This study tests two hypotheses related to body color, activity period, and eye structure. As detailed above, pale species tend to be restricted to lightless and dim light environments. Consequently, we first test the prediction that in four genera of ants activity period of pale species likewise is restricted to dim light conditions, i.e., nocturnal, whereas activity period of their dark congeners is not restricted to dim light conditions. To this end, we first quantified for each species the association between cuticular pigmentation and daily above-ground activity patterns, i.e., whether pale species are more likely to be nocturnal than their dark congeners. Second, we tested the prediction derived from the arguments above that eye structure varies with body color and activity time. Specifically, we examined whether within each genus pale species have larger eyes than their dark congeners, i.e., eyes that would enhance vision in dim

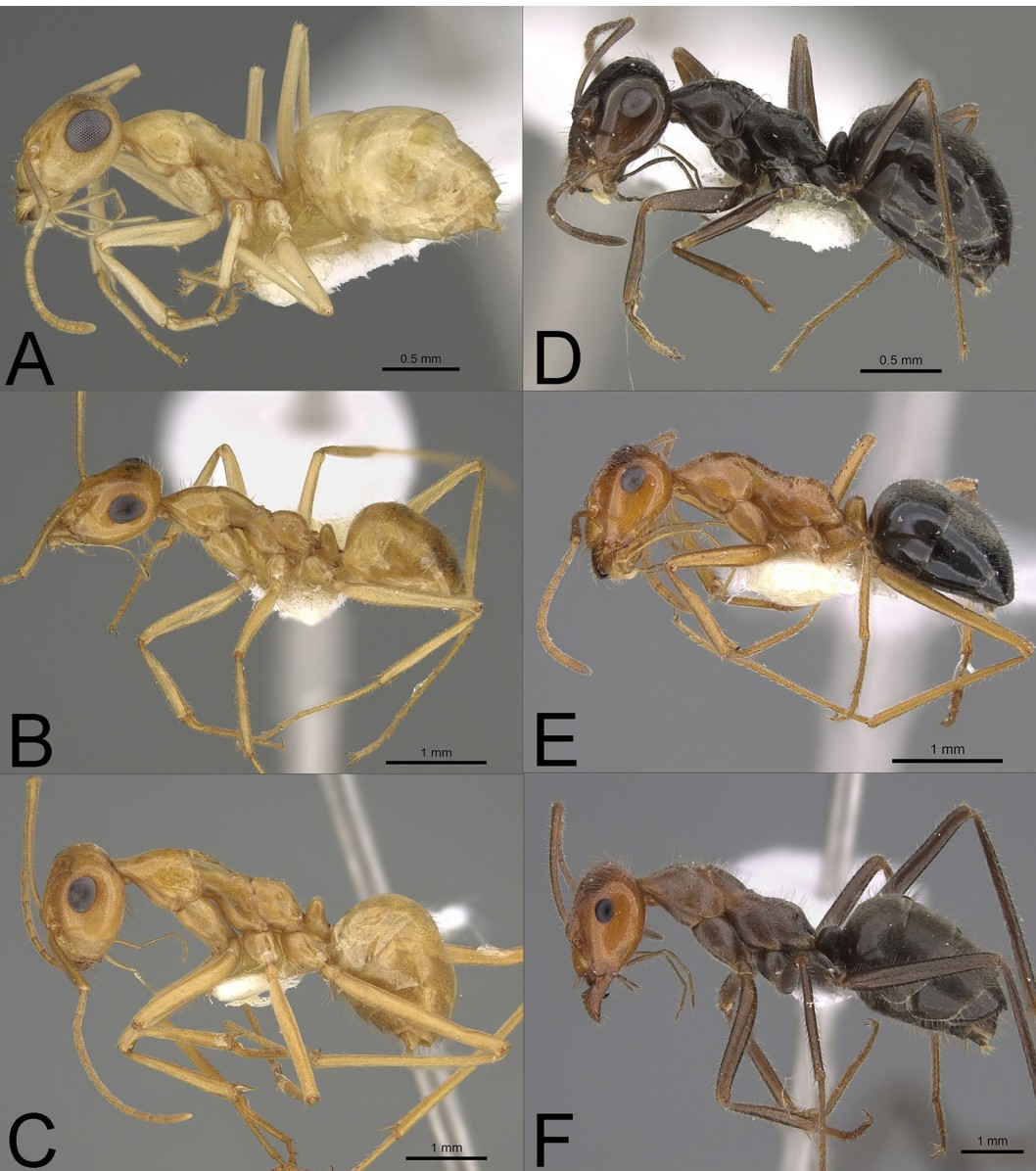

**Fig 1.** Profile photographs of pale **(A-C)** and dark species **(D-F)** (see text) of *Myrmecocystus* (subfamily Formicinae) examined in this study: (**A**) *M. christineae* (CASENT0923358), (**B**) *M. navajo* (CASENT0923356), (**C**) *M. mexicanus*-02 (CASENT0923355), (**D**) *M. yuma* (CASENT0923359), (**E**) *M. kennedyi* (CASENT0923362), and (**F**) *M. mendax*-03 (ASUSIBR00001132). Note the relatively larger eyes of pale compared to dark species. Species are arranged by size pairs–**A&D**, **B&E**, and **C&F** (see text). Photographs by Michele Esposito from www.antweb.org.

conditions. Several studies have compared the compound eyes of nocturnal and diurnal ants [29]. However, most of these comparative studies lacked adequate controls for phylogeny in that they compared relatively small numbers of species from different lineages or with unknown phylogenetic relationships [29, 31, 32]. In this study, phylogenies were available for all four examined ant genera, such that we could compare eye structure of closely related pale and dark species. However, phylogenetic corrections were not performed.

For a subset of these species, we examined eye structure in more detail to explore the effects of activity-pattern-related variation in eye size on visual sensitivity, acuity, and field

dimensions. High visual acuity is generally expected for members of the Hymenoptera (i.e., ants, wasps, bees) whose apposition eyes [33] are typically structured to maximize image resolution rather than light capture. Visual resolution of apposition eyes can be assessed by measuring facet diameter ($D$) and interommatidial angle ($\Delta\phi$, the angle between the optical axes of adjacent ommatidia). Larger facets capture more light and $D$ is positively correlated with sensitivity, while resolving power is negatively correlated with $\Delta\phi$. Consequently, there may be a potential tradeoff between resolution and sensitivity given that $D$, $\Delta\phi$, and eye size interact in complex and sometimes opposing ways [34]. Nocturnal animals usually resolve this tradeoff in favor of sensitivity, and thus have lower acuity compared to their diurnal counterparts [35].

We also used the eye parameter ($\rho$, which is the product of $D$ in μm and $\Delta\phi$ in radians) [35–37], to characterize the compromise between sensitivity and acuity for each species. The calculated $\rho$ indicates how closely the eye develops to the limits imposed by diffraction, i.e., whether the eye is structured to enhance resolution over sensitivity. These values are low for diurnally active species, and they increase at lower light intensities with nocturnal species often having $\rho$ values $\geq 2$.

For two species within each genus (one pale, one dark), we also measured $\Delta\phi$ in the center of the eye, visual field span, and regional variation in $D$. Collectively, these measures permit inferences about how visual field structure varies between nocturnal and diurnal species both within and across genera.

We also examined variation in size of the ocelli in *Myrmecocystus*, which was the only examined genus in which workers possess these structures. Ocelli are a second visual system present in most flying insects that detect polarized light and assist in head stabilization and horizon detection [38–40], but also reflect the natural history and environment of the species [41]. Ocelli are present in alate queens and males of nearly all ant species, but they typically are lacking in the pedestrian workers with *Myrmecocystus* being a notable exception [42, 43]. Snelling [30] noted that the ocelli were smaller in pale than in dark species of *Myrmecocystus*.

All species examined herein were geographically restricted to the southwestern United States and northwestern Mexico. The relatively large number of pale species found in this region suggests that pale ants may occur in other regions of the world and in other genera and/or subfamilies. Consequently, we assessed variation in ant color and its correlation with eye size on a larger geographical and taxonomic scale by surveying photographs across species in several ant genera. We could not test the correlation between body color and nocturnal activity in these species, but we expect pale body color to be correlated with eye morphology that enhances visual sensitivity.

## Methods

### Study species

We examined the relationship between body color, activity pattern, and eye structure in 23 ant species spread across four genera in two subfamilies–*Myrmecocystus* (subfamily Formicinae) and *Aphaenogaster*, *Temnothorax*, and *Veromessor* (subfamily Myrmicinae); all four genera contained pale and dark species. Hereafter, names of pale species are in normal font, and names of dark species in **bold** font.

*Myrmecocystus*: We examined 74 workers from six species (Fig 1). This genus is restricted to North America, and it consists of 29 described species [30, 44], plus several undescribed and cryptic species [45]. Species of *Myrmecocystus* display a wide range of body sizes, and all species are size polymorphic. We compared three size-similar species pairs that differed in pigmentation across this range of body sizes: small (*M. christineae* and ***M. yuma***), medium (*M. navajo* and ***M. kennedyi***), and large (*M. mexicanus*-02 and ***M. mendax*-03**). The genus consists

of two major clades: most pale species occur in one clade, and the rest of the pale species and all dark species occur in the other clade. The latter clade consists of six smaller clades–one includes the rest of the pale species, and the other five consist of dark species [45]. Consequently, we could not compare eye structure for phylogenetically adjacent pale and dark species. Instead, we chose both pale and dark species pairs based on a combination of availability of specimens, taxonomic stability, and relative ease of identification.

In his revision of *Myrmecocystus*, Snelling [30] also indicated that the ocelli were reduced in size or absent in pale species compared to their dark congeners. We tested this observation by measuring diameter of the anterior ocellus for workers of the above six species, plus workers of three additional pale species (*M. ewarti*, *M. testaceus*, *M. mexicanus*-01). As such, our analysis included all known pale species except *M. melanoticus* and *M. pyramicus* [30, 45], for which specimens were not available.

*Aphaenogaster*: We examined 38 workers from four species (Fig 2). This genus includes 30 species in North America. We compared *A. megommata*, the only pale species, with three closely related dark congeners **A. boulderensis**, **A. occidentalis**, and **A. patruelis** [46, 47].

*Temnothorax*: We examined 29 workers from three species (Fig 3). In North America, this genus consists of more than 80 described species plus numerous undescribed species, with the *T. silvestrii* clade consisting of several poorly known, poorly collected pale species (M. Prebus, pers. comm.). Our analysis included the undescribed pale species *T*. sp. BCA-5 [in 48, as *Leptothorax* sp. BCA-5] from the *T. silvestrii* group, and **T. neomexicanus** and **T. tricarinatus**, which are two dark species from the *T. tricarinatus* group, which is sister to the *T. silvestrii* group [49, M. Prebus, pers. comm.].

*Veromessor*: We examined 133 workers from the 10 species that occur in the genus (Fig 4). This genus only occurs in North America [50, 51]; two species are pale, *V. lariversi* and *V*. RAJ-*pseu*, while the eight other species are dark.

All specimens are deposited in the collections of Robert. A. Johnson, Tempe, AZ (RAJC), and Matthew M. Prebus, Tempe, AZ (MMPC).

## Measure of body coloration

All brightness, body size, and eye measurements were made from photographs of workers as described below. Brightness (B) was measured using the color window in Adobe Photoshop from photographs downloaded from Antweb (www.antweb.org). This assessment is similar to the lightness value in HSL that has been used to characterize body color in other studies of ants [10, 12]. We excluded specimens that were obviously discolored, i.e., those in which the color differed substantially from intraspecific specimens recently collected by RAJ. Using the photograph of the worker body in profile, we measured the average B over an $11 \times 11$ pixel area (ca. 8.2 mm$^2$) on the head (immediately posterior to the eye), mesosoma (center of mesopleura), and gaster (anteroposterior portion of first gastral tergum), then averaged these values for each worker, then averaged that value across all workers for each species. In measuring B, we focused on areas of diffuse reflectance and avoided spots of specular (mirror-like) reflectance so as to minimize their effect on B values. Using the average B value across the three body tagmata accounted for intra-individual variation in color that occurs in many ant species. All measurements were made by RAJ. We compared using a t-test mean B values for species qualitatively categorized as pale versus dark.

## Activity patterns

The relationship between color and activity pattern was evaluated by gleaning above-ground foraging times from literature, personal observations, and personal communications. Foraging

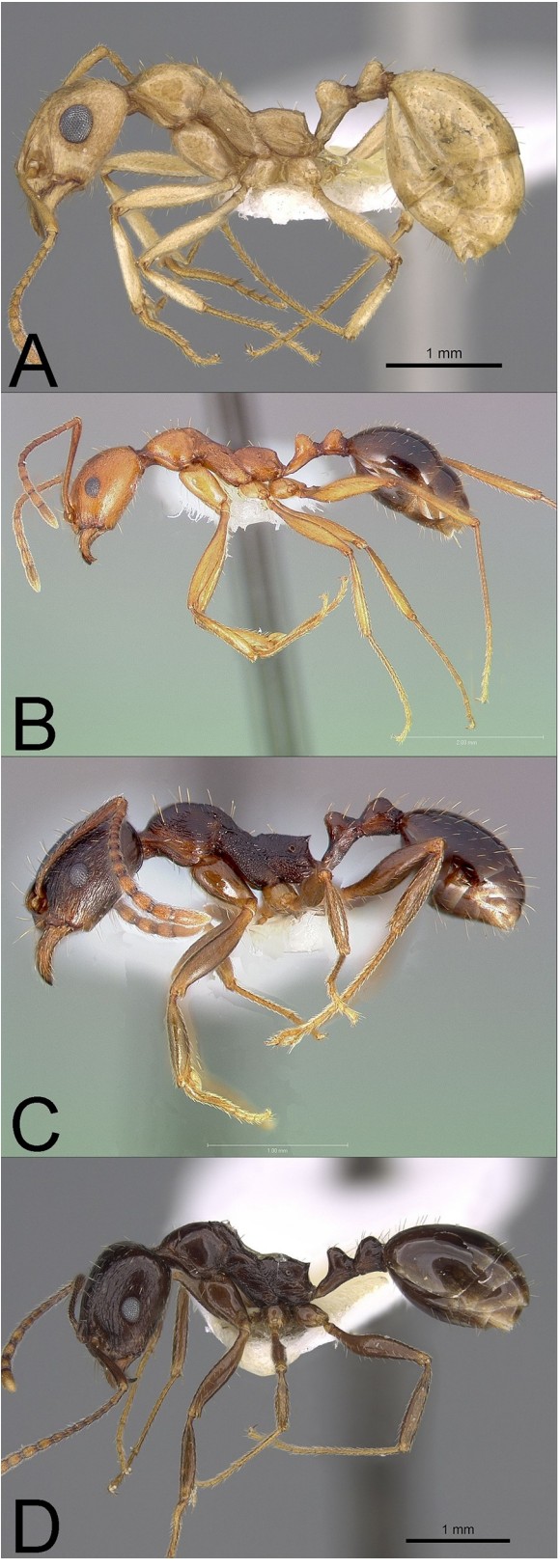

**Fig 2.** Profile photographs of pale **(A)** and dark species **(B-D)** (see text) of *Aphaenogaster* (subfamily Myrmicinae) examined in this study: **(A)** *A. megommata* (CASENT0923367), **(B)** *A. boulderensis* (CASENT0005722), **(C)** *A.*

*occidentalis* (CASENT0005725), and (**D**) *A. patruelis* (CASENT0923366). Note the relatively larger eyes of pale compared to dark species. Photographs by Michele Esposito from www.antweb.org.

times were classified as one or more of the following: diurnal, nocturnal, matinal, crepuscular, and variable. Exclusively day-active or night-active species were classified as diurnal and nocturnal, respectively. The category "variable" included species that forage during both day and night and those that have matinal and crepuscular foraging. This includes species in which foraging time varies seasonally with temperature–diurnal when days are cool, crepuscular-matinal as temperatures increase, and nocturnal when nights are warm. We tested the association between color (pale and dark) and activity time (diurnal, nocturnal, variable) using a Fisher's exact test [52].

## Body size and eye measurements

We measured body size and eye characteristics for workers from all 23 species listed above. These measurements included up to 15 workers per species, selected from up to seven colonies (when available), so as to include variation within and across colonies. Workers of several species display weak to strong size polymorphism, and we selected workers within colonies of these species to span the variation in body size. Body size was measured as mesosoma length (= Weber's length), which is a standard measure for body size in ants [53]. Mesosoma length was measured as the diagonal length of the mesosoma in profile from the point at which the pronotum meets the cervical shield to the posterior base of the metapleural lobe. Mesosoma length was measured from photographs taken using a Spot Insight QE camera attached to a Leica MZ $12_5$ microscope. Images were then displayed on a video monitor, and mesosoma length was measured using ImageJ (available at http://rsb.info.nih.gov/nih-image/). Measurements were calibrated using photographs of an ocular micrometer scaled in 0.01 mm increments.

Eye measurements were made from high-resolution photographs of the left eye taken in profile focused on the center of the eye at an angle that allowed viewing all facets. Photographs were taken using a Leica M205C microscope at 100× that was linked to the stacking software program Helicon Focus (www.heliconsoft.com/heliconsoft-products/helicon-focus). This software combines photographs taken in different focus planes into one photograph where the entire eye surface is in focus. Facet lenses were counted, and eye area and facet diameter (*D*) were measured using Digimizer (https://www.digimizer.com/). The area tool was used to calculate area. This tool also calculated the centroid of the eye, and *D* was the average of four adjoining facets at the centroid. We also measured diameter of the anterior ocellus for species of *Myrmecocystus*. All photographs contained a 0.15 mm scale bar for calibrating measurements.

## Detailed eye measurements

Detailed measures of eyes and visual field were taken from a subset of the 23 species. We conducted four eye measurements on two species from each genus, one pale and the other dark, which were closely matched in mesosoma length.

**Interommatidial angle ($\Delta\phi$), eye parameter ($\rho$), and visual field.** Measurements of $\Delta\phi$ allowed us to examine how spatial resolution varied with activity period, eye size, and body size. We measured $\Delta\phi$ in the lateral region of the eye for five workers from one pale and one dark species in each of the four genera (*M. navajo*, ***M. kennedyi*** in *Myrmecocystus*; *A. megommata*, ***A. occidentalis*** in *Aphaenogaster*; *T.* BCA-5, ***T. neomexicanus*** in *Temnothorax*; *V.*

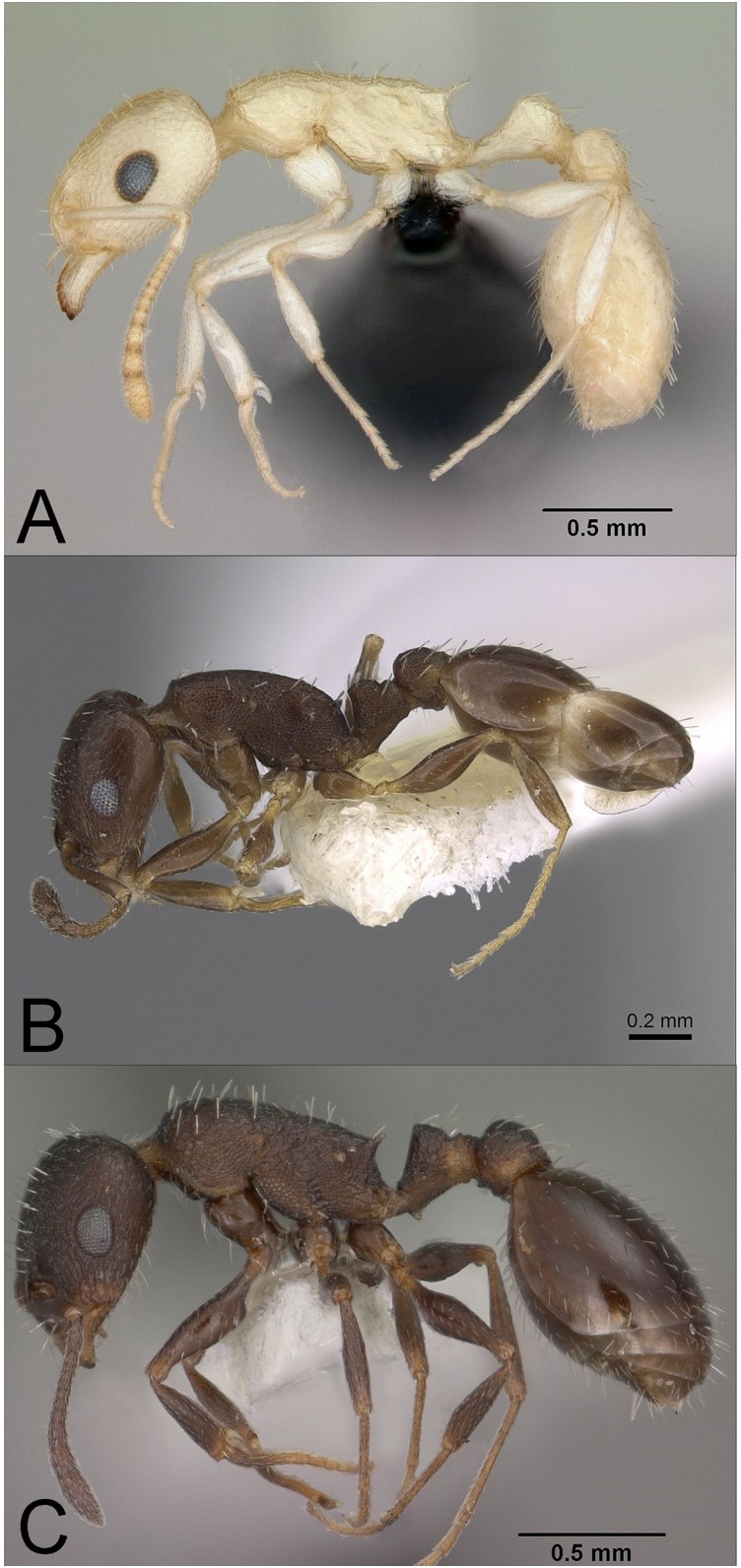

**Fig 3.** Profile photographs of pale **(A)** and dark species **(B–C)** (see text) of *Temnothorax* (subfamily Myrmicinae) examined in this study: **(A)** *T.* sp. BCA-5 (CASENT0118165), **(B)** *T. neomexicanus* (CASENT0923368), and **(C)** *T. tricarinatus* (CASENT0102845). Note the relatively larger eyes of pale compared to dark species. Photographs by Michele Esposito from www.antweb.org.

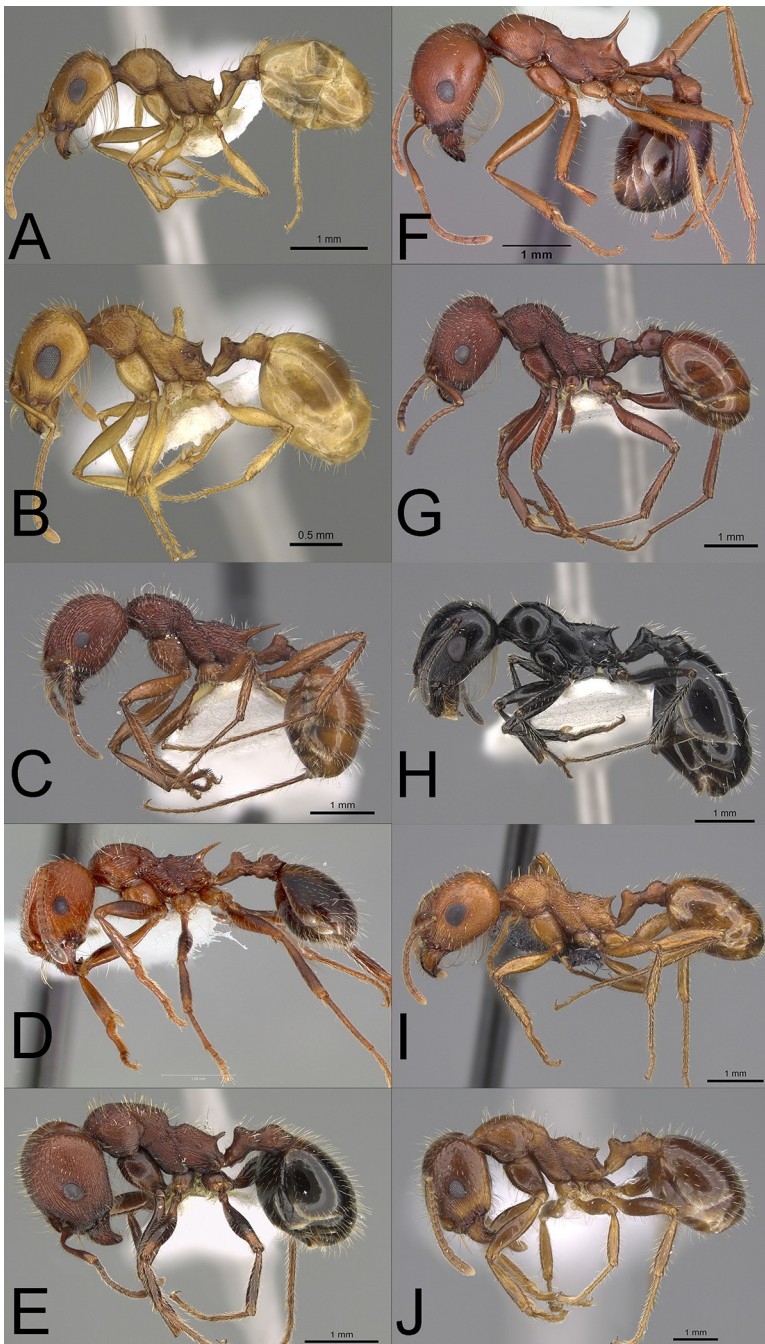

**Fig 4. Profile photographs of pale (A–B) and dark species (C–J)** (see text) of *Veromessor* (subfamily Myrmicinae) examined in this study: (**A**) *V. lariversi* (CASENT0923345), (**B**) *V.* RAJ-*pseu* (CASENT0923346), (**C**) *V. andrei* (CASENT0923137), (**D**) *V. chamberlini* (CASENT0005730), (**E**) *V. chicoensis* (CASENT0923347), (**F**) *V. julianus* (CASENT0104946), (**G**) *V. lobognathus* (CASENT0923126), (**H**) *V. pergandei* (CASENT0923124, (**I**) *V. smithi* (MCZ-ENT00671466), and (**J**) *V. stoddardi* (CASENT0922825). Note the relatively larger eyes of pale compared to dark species. Photographs by Wade Lee, April Nobile, and Michele Esposito from www.antweb.org.

*lariversi*, **V. chicoensis** in *Veromessor*) using the radius of curvature method outlined in Bergman and Rutowski [54] with minor modifications. For each specimen, we photographed the left eye in side view from a position on a line perpendicular to the anterior-posterior axis of

the eye. This created a photograph showing the edge of the eye surface corresponding to the lateral region with individual facets visible at the eye edge. Each image was copied into Geogebra (©International Geogebra Institute, 2013; www.geogebra.org) to measure the angle subtended by the eye surface spanned by two facets in the apical region. Briefly, a point was identified between two facets at the edge of the eye at its apex. We then drew a line to a point on the eye surface two facet rows away in the anterior direction. This was taken as the tangent to the eye at that point and the perpendicular bisector of these lines between these points was drawn. This was also done for a line extending from the original point to a point two facet rows in the posterior direction. The $\Delta\phi$ was the angle between these two perpendicular bisectors divided by two. We measured $\Delta\phi$ three times in the same area for each worker and used the average in our analysis. We calculated $\rho$ for each worker as the product of $\Delta\phi$ in radians and $D$ in μm.

The same images that were used to measure $\Delta\phi$ also were used to measure the anterior-posterior visual field span. Lines perpendicular to the tangent of the eye surface were drawn at the anterior and posterior edge of each eye. The angular span of the visual field along the anterior-posterior axis was characterized by the angle between these lines. This measurement was repeated three times for each specimen and the mean value was used in our analyses.

**Regional variation in $D$.**   Regional variation in $D$ was measured for *Myrmecocystus* and *Veromessor*. The small eyes of *Aphaenogaster* and *Temnothorax* contained too few facets to warrant examination of regional variation. We quantified regional variation in $D$ using the photographs taken for eye size measurements (see above) for one pale and one dark species of *Myrmecocystus* (*M. navajo* and **M. kennedyi**) and *Veromessor* (*V. lariversi* and **V. chicocensis**). The image for each individual was printed on letter size paper, and $D$'s were measured in five eye regions: anterior, dorsal, lateral, posterior, and ventral. The anterior-posterior axis of the eye was a line from the mandible to the posterior corner of the head, and eye regions were described relative to this line. In each region, three facets in one row were measured to the nearest 0.1 mm with digital calipers, then scaled using the 0.15 mm scale bar present on each photograph. Mean $D$ in each region was total length divided by three.

## Data analysis

Eye area, facet number, and mean $D$ (dependent variables) were compared across species (independent variable) within each genus using a multivariate analysis-of-covariance (MANCOVA) in the general linear models (GLM) program of SPSS [52, 55, 56]; mesosoma length was the covariate. A least significant difference (LSD) post-hoc test compared the estimated marginal means across species for each variable using mesosoma length as a covariate. Diameter of the anterior ocellus (dependent variable) was compared similarly across species (independent variable) of *Myrmecocystus* using analysis-of-covariance (ANCOVA), with mesosoma length included as the covariate. *Myrmecocystus navajo* was omitted from this analysis because only one worker had a visible anterior ocellus (see below).

The dependent variables–$\Delta\phi$, $\rho$, and visual field span—were compared in separate ANCOVA's using genus (4 levels) and activity period (diurnal versus nocturnal) as independent variables, with mesosoma length as a covariate [52]. A Tukey's HSD post-hoc test compared differences across genera and species for each variable. Regional variation in $D$ was analyzed using a one-way repeated measures ANOVA followed by a LSD post-hoc test for each species [52, 57]. For all tests, data were transformed, as necessary, to meet the assumptions for homogeneity of variance-covariance matrices (Box's M test and Levene's test) and homogeneity of regression slopes [56, 58].

### Survey for additional pale ant species

We used Antweb (www.antweb.org) to scan photographs for pale ant species in the genera *Aphaenogaster*, *Crematogaster*, *Messor*, and *Temnothorax* (subfamily Myrmicinae), and *Dorymyrmex* and *Iridomyrmex* (subfamily Dolichoderinae). We scrolled through frontal photographs of the head for all species in each genus looking for species that appeared pale. We verified our visual assessment of color for these taxa by measuring their brightness value (B) using Adobe Photoshop, as detailed above, with pale species being those with a mean B value > 70. We then assessed relative eye size for these species, calculated as eye area/mesosoma length, compared to their dark congeners, which all had B values < 70. Phylogenies were not available for these genera, so our dark species comprised the first five species/nominal subspecies on the Antweb page from each genus. Pale species with enlarged eyes were those that had a relative eye size larger than that of all five dark congeners. Photographs of each species can be viewed by going to https://www.antweb.org/advSearch.do, then placing the genus and species name in the advanced search box.

## Results

### Pigmentation and daily activity pattern

We first quantitatively confirmed our visual impressions of variation in body color. As predicted, our brightness values (B) measured from Antweb photographs were consistently higher for species that we visually classified as pale compared to dark across all four genera (t-test, $t = -10.1_{24 df}$, $P < 0.0001$). Mean B values were 74.6 (range 65.1–87.8, $n = 10$) for pale species and 45.4 (range 32.3–58.7, $n = 16$) for dark species (Table 1). Values did not overlap for any pale and dark species as all dark species had a mean B below 60, whereas all pale species had a mean B above 65. However, note that the two pale species of *Veromessor* displayed mean B values that ranged from 65–70, which was intermediate to pale and dark species in the other three genera (Table 1). There also was a significant effect of color (pale, dark) on activity period (diurnal, nocturnal, variable) ($P < 0.0001$, Fisher's exact test) with a preponderance of pale species that are nocturnal and dark species that have diurnal/variable activity periods (Table 2). Henceforth, we use pale and dark to refer to species with nocturnal and diurnal/variable activity periods, respectively.

### Eye area, facet number, and facet diameter

We determined the magnitude of interspecific differences in eye area, facet number, and facet diameter for species in each of the four genera. Before running the MANCOVA for each genus, we tested the homogeneity of variance-covariance (Box's M test and Levene's test) and homogeneity of slopes assumptions (Wilks' lambda for the species × mesosoma interaction effect). Box's M test for homogeneity of variance-covariance matrices is not robust to violations of this assumption such that an alpha level of $P < 0.001$ is recommended [69, 70]. *Myrmecocystus* and *Aphaenogaster* met both assumptions. *Temnothorax* met the homogeneity of variance-covariance assumption but not the assumption for homogeneity of regression slopes because the slopes for eye area differed across species ($P = 0.007$) (Table 3). Consequently, we reran the model after deleting eye area, and the homogeneity of regression slopes assumption was met using the two other dependent variables (facet number, facet diameter) (Table 3). We then analyzed variation in eye area across species separately using a one-way ANOVA. The dependent variable in this model was relative eye size, calculated as eye area/mesosoma length, which standardized eye area for body size (see above). *Veromessor* did not meet the homogeneity of variance-covariance assumption, so we used Pillai's trace instead of Wilks'lambda to

**Table 1. Brightness values, foraging time, and the four morphological traits measured for ant species examined in this study.** Brightness values are given as mean ± 1 SE (number of workers) (see text). Morphological measures are given as mean (± 1 SE) (also see Figs 5 & 7–9); for the three eye traits, values on the first line are raw data, values on second line are estimated marginal mean values using mesosoma length as a covariate in a multivariate analysis-of-covariance. Species are listed alphabetically by subfamily, genus, and species within each genus. Taxonomy follows Bolton [59]; species followed by a number are undescribed or in the process of revision [see 45]. Pale species in normal font; dark species in **bold** font (see text).

| Species | Brightness value | Foraging time | References | Morphological measures | | | |
|---|---|---|---|---|---|---|---|
| | | | | Eye area (mm²) | Number of eye facets | Facet diameter (D) (μm) | Mesosoma length (mm) |
| **Subfamily Formicidae *Myrmecocystus*[a]** | | | | | | | |
| *christineae* Snelling | 75.2 ± 2.4 (4) | nocturnal* | [60] | 0.0763 ± 0.0049 | 277.5 ± 9.3 | 19.68 + 0.26 | 1.14 ± 0.04 |
| | | | | 0.1196 ± 0.0077 | 384.1 ± 14.2 | 20.81 ± 0.49 | |
| *ewarti* Snelling[#] | 72.7 ± 1.7 (3) | nocturnal | [30, 60] | | | | |
| ***kennedyi* Snelling** | 55.9 ± 0.2 (3) | diurnal | [30, 60] | 0.0463 ± 0.0025 | 322.8 ± 11.3 | 13.71 ± 0.22 | 1.43 ± 0.04 |
| | | | | 0.0685 ± 0.0058 | 377.6 ± 10.7 | 14.29 ± 0.37 | |
| **sp. *mendax*-03** | 58.7 ± 4.5 (3) | diurnal | [30] | 0.0817 ± 0.0048 | 457.4 ± 16.3 | 15.85 ± 0.26 | 2.31 ± 0.10 |
| | | | | 0.0413 ± 0.0067 | 358.2 ± 12.4 | 14.79 ± 0.43 | |
| sp. *mexicanus*-01[#] | 73.8 + 1.1 (4) | nocturnal | [30, 60] | | | | |
| sp. *mexicanus*-02 | 78.4 ± 4.5 (3) | nocturnal | [30, 60] | 0.1609 ± 0.0118 | 524.1 ± 21.3 | 21.83 ± 0.50 | 2.31 ± 0.08 |
| | | | | 0.1207 ± 0.0067 | 425.1 ± 12.4 | 20.78 ± 0.43 | |
| *navajo* Wheeler | 74.1 + 2.8 (6) | nocturnal, crepuscular in cooler months | [30, 60] | 0.1052 ± 0.0068 | 349.8 ± 14.5 | 20.27 ± 0.37 | 1.67 ± 0.07 |
| | | | | 0.1103 ± 0.0052 | 352.3 ± 9.6 | 20.41 ± 0.33 | |
| *testaceus* Emery[#] | 72.3 ± 1.7 (3) | nocturnal | [30, 60] | | | | |
| ***yuma* Wheeler** | 38.7 ± 7.1 (3) | matinal-crepuscular | [30] | 0.0378 ± 0.0009 | 287.0 ± 4.2 | 13.25 ± 0.21 | 1.12 ± 0.02 |
| | | | | 0.0825 ± 0.0079 | 397.1 ± 14.4 | 14.42 ± 0.50 | |
| **Subfamily Myrmicinae *Aphaenogaster*[b]** | | | | | | | |
| ***boulderensis* Smith** | 53.7 ± 8.0 (2) | crepuscular, nocturnal, matinal | [60] | 0.0480 ± 0.0000 | 147.5 ± 3.5 | 19.88 ± 0.38 | 2.14 ± 0.02 |
| | | | | 0.0176 ± 0.0067 | 81.3 ± 18.7 | 18.49 ± 1.03 | |
| *megommata* Smith | 76.9 ± 2.9 (6) | crepuscular, nocturnal | [60, 61] | 0.0815 ± 0.0032 | 241.4 ± 7.9 | 23.10 ± 0.23 | 1.68 ± 0.03 |
| | | | | 0.0745 ± 0.0020 | 226.1 ± 5.7 | 22.78 ± 0.31 | |
| ***occidentalis* (Emery)** | 46.7 ± 3.4 (6) | variable[+] | B. DeMarco, pers. comm.; P. S. Ward, pers. comm. | 0.0264 ± 0.0008 | 79.8 ± 2.1 | 19.58 ± 0.26 | 1.42 ± 0.03 |
| | | | | 0.0328 ± 0.0019 | 93.7 ± 5.4 | 19.87 ± 0.30 | |
| ***patruelis* Forel** | 42.4 ± 4.2 (9) | variable | D. Holway, pers. comm.; P.S. Ward, pers. comm. | 0.0331 ± 0.0019 | 104.1 ± 4.9 | 19.96 ± 0.26 | 1.44 ± 0.02 |
| | | | | 0.0388 ± 0.0019 | 116.6 ± 5.3 | 20.22 ± 0.29 | |
| **Subfamily Myrmicinae *Temnothorax*[c]** | | | | | | | |
| sp. BCA-5 | 87.3 ± 0.2 (2) | nocturnal | [48, as *Leptothorax* sp. BCA-5]; R.A. Johnson, pers. obs. | 0.0238 ± 0.0026 | 92.2 ± 6.9 | 18.30 ± 0.29 | 0.93 ± 0.06 |
| | | | | 0.0201 ± 0.0011 | 80.8 ± 3.7 | 17.76 ± 0.39 | |
| ***neomexicanus* (Wheeeler)** | 35.4 ± 2.8 (4) | crepuscular | S.P. Cover, pers. comm. | 0.0109 ± 0.0002 | 66.3 ± 0.9 | 13.17 ± 0.16 | 0.67 ± 0.02 |
| | | | | 0.0128 ± 0.0006 | 72.0 ± 2.1 | 14.43 ± 0.23 | |
| ***tricarinatus* (Emery)** | 39.3 ± 3.2 (3) | crepuscular | S.P. Cover, pers. comm. | 0.0138 ± 0.0003 | 89.0 ± 2.1 | 13.46 ± 0.23 | 0.77 ± 0.02 |
| | | | | 0.0135 ± 0.0005 | 88.1 ± 1.7 | 13.42 ± 0.10 | |
| **Subfamily Myrmicinae *Veromessor*[d]** | | | | | | | |
| ***andrei* (Mayr)** | 40.1 ± 2.2 (15) | variable | [62, 63] | 0.0663 ± 0.0045 | 198.4 ± 9.1 | 21.00 ± 0.29 | 2.14 ± 0.06 |
| | | | | 0.0492 ± 0.0022 | 170.5 ± 4.9 | 20.27 ± 0.29 | |
| ***chamberlini* (Wheeler)** | 47.9 ± 4.2 (6) | diurnal | M. Bennett, pers. comm.; R. A. Johnson, pers. obs. | 0.0395 ± 0.0013 | 126.7 ± 3.2 | 19.60 ± 0.23 | 1.62 ± 0.03 |
| | | | | 0.0513 ± 0.0021 | 145.9 ± 4.7 | 20.11 ± 0.28 | |
| ***chicoensis* Smith** | 44.6 ± 2.1 (6) | variable | [64] | 0.0678 ± 0.0062 | 167.6 ± 10.1 | 21.00 ± 0.26 | 1.92 ± 0.09 |
| | | | | 0.0629 ± 0.0019 | 159.6 ± 4.2 | 20.79 ± 0.25 | |
| ***julianus* (Pergande)** | 46.1 ± 4.1 (6) | crepuscular-nocturnal-matinal | [65] | 0.0771 ± 0.0038 | 198.8 ± 7.7 | 22.07 ± 0.27 | 2.09 ± 0.06 |
| | | | | 0.0629 ± 0.0020 | 175.5 ± 4.5 | 21.46 ± 0.26 | |

(*Continued*)

**Table 1.** (Continued)

| Species | Brightness value | Foraging time | References | Morphological measures | | | |
|---|---|---|---|---|---|---|---|
| | | | | Eye area (mm$^2$) | Number of eye facets | Facet diameter (D) (μm) | Mesosoma length (mm) |
| *lariversi* Smith | 68.5 ± 1.6 (9) | nocturnal | R.A. Johnson, pers. obs. | 0.0796 ± 0.0027 | 223.5 ± 4.9 | 21.00 ± 0.31 | 1.61 ± 0.05 |
| | | | | 0.0920± 0.0019 | 243.7 ± 4.4 | 21.53 ± 0.26 | |
| *logobnathus* (Andrews) | 50.4 ± 5.0 (6) | variable | [66, 67]; M. Bennett, pers. comm. | 0.0808 ± 0.0024 | 223.8 ± 2.4 | 20.98 ± 0.33 | 1.94 ± 0.05 |
| | | | | 0.0752 ± 0.0020 | 214.7 ± 4.5 | 20.74 ± 0.27 | |
| *pergandei* (Mayr) | 32.3 ± 1.8 (4) | variable | [60, 67]; R.A. Johnson, pers. obs. | 0.0805 ± 0.0049 | 231.9 ± 8.8 | 19.87 ± 0.26 | 1.90 ± 0.06 |
| | | | | 0.0771 ± 0.0018 | 226.3 ± 4.0 | 19.72 ± 0.24 | |
| RAJ-*pseu* | 65.1 ± 1.6 (8) | nocturnal | [67, as *V. lariversi*]; R.A. Johnson, pers. obs. | 0.0910 ± 0.0016 | 250.6 ± 3.6 | 21.17 ± 0.26 | 1.47 ± 0.18 |
| | | | | 0.1112 ± 0.0022 | 283.7 ± 4.9 | 22.04 ± 0.29 | |
| *smithi* Cole | 55.5 ± 3.9 (6) | crepuscular-nocturnal | [67, 68]; M. Bennett, pers. comm. | 0.1010 ± 0.0035 | 240.1 ± 5.8 | 22.52 ± 0.32 | 1.79 ± 0.04 |
| | | | | 0.1032 ± 0.0020 | 243.6 ± 4.5 | 22.61 ± 0.26 | |
| *stoddardi* (Emery) | 42.9 ± 1.2 (4) | crepuscular-nocturnal | M. Bennett, pers. comm. | 0.0483 ± 0.0035 | 130.7 ± 6.4 | 20.43 ± 0.31 | 1.85 ± 0.08 |
| | | | | 0.0476 ± 0.0018 | 129.5 ± 4.0 | 20.40 ± 0.24 | |

[#] only ocelli measured.

[*] foragers of pale species are nocturnal, except that they sometimes exit nests just prior to dusk, and sometimes forage on overcast days.

[+] foraging time varies seasonally depending on temperature–diurnal when days are cool, crepuscular-matinal as temperatures increase, nocturnal when nights are warm.

[a] = estimated marginal means evaluated at a mesosoma length of 1.7434 mm.

[b] = estimated marginal means evaluated at a mesosoma length of 1.5468 mm.

[c] = estimated marginal means evaluated at a mesosoma length of 0.7534 mm.

[d] = estimated marginal means evaluated at a mesosoma length of 1.8333 mm.

evaluate multivariate significance [71]. The Levene's test for *Veromessor* was also significant for the regression slopes comparison and in the final model. Consequently, we decreased the *P* value for post-hoc tests to $P < 0.01$ [70].

**Myrmecocystus.** Eye structure (eye area, facet number, facet diameter) differed among species of *Myrmecocystus* after controlling for mesosoma length (Wilks' λ = 0.031, $F_{15,180}$ = 30.0, $P < 0.001$). The tests of between-subject effects demonstrated that all three dependent variables differed across species (eye area: $F_{5,67}$ = 76.4, $P < 0.001$; facet number: $F_{5,67}$ = 4.9, $P < 0.001$; mean *D*: $F_{5,67}$ = 127.5, $P < 0.001$; Fig 5). Based on the estimated marginal means, pairwise comparisons across all species pairs using a LSD test showed that eye area and mean *D* were larger in pale species (*M. christineae*, *M. navajo*, *M. mexicanus*-02) than in their dark congeners (**M. yuma**, **M. kennedyi**, **M. mendax-03**) ($P < 0.001$, Fig 5). Facet number varied among species in a different pattern, with this number higher in *M. mexicanus*-02 and **M. yuma**, but with **M. yuma** also overlapping the four other congeners with fewer facets. Relative

**Table 2. Association between color and activity period based on data in Table 1.** Pale species have a brightness (B) value > 65 and dark species have a B value < 60 as measured in Adobe Photoshop (see text).

| Cuticular coloration | Foraging time | | | |
|---|---|---|---|---|
| | Nocturnal | Variable[*] | Diurnal | Total |
| **Pale** | 9 | 1 | 0 | 10 |
| **Dark** | 0 | 13 | 3 | 16 |
| **Total** | 9 | 14 | 3 | 26 |

[*] variable includes species in which foraging time varies seasonally and those that display crepuscular-matinal activity.

**Table 3. Results for the homogeneity of variance-covariance and regression slopes assumptions in the MANCOVA.** The MANCOVA for each genus was run twice; the first run tested homogeneity of variance-covariance and homogeneity of regression slopes assumptions, and the second run was for results of the model. For the Levene's test, the first column gives the $P$ value for the assumptions run, the second gives the $P$ value for the results run. Values for these three lines are eye area, facet number, and facet diameter, respectively. For *Temnothorax*, the first line is for all three variables, the second line is the model after deleting eye area (see text); values in the second line for the Levene's test are for facet number and facet diameter, respectively.

| Genus | Homogeneity of variance-covariance tests | | | | | Homogeneity of regression slopes test | | |
|---|---|---|---|---|---|---|---|---|
| | Box's M test | *P* | Levene's test | | | Wilks' λ | F | *P* |
| | | | Assumptions | Results | | | | |
| *Myrmecocystus* | $F_{30, 8480} = 1.70$ | 0.010 | 0.017 | 0.068 | | 0.721 | $F_{15, 166} = 1.40$ | 0.155 |
| | | | 0.048 | 0.38 | | | | |
| | | | 0.074 | 0.078 | | | | |
| *Aphaenogaster* | $F_{12, 5277} = 0.98$ | 0.47 | 0.20 | 0.12 | | 0.743 | $F_{9, 68} = 0.99$ | 0.46 |
| | | | 0.38 | 0.21 | | | | |
| | | | 0.28 | 0.89 | | | | |
| *Temnothorax* | $F_{12, 728} = 2.25$ | 0.009 | 0.62 | 0.18 | | 0.526 | $F_{6, 42} = 2.66$ | 0.028 |
| | | | 0.42 | 0.60 | | | | |
| | | | 0.55 | 0.55 | | | | |
| | $F_{54, 1410} = 2.21$ | 0.040 | 0.42 | 0.86 | | 0.666 | $F_{4, 44} = 2.48$ | 0.057 |
| | | | 0.55 | 0.55 | | | | |
| *Veromessor* | $F_{54, 18615} = 1.80$ | 0.0003* | 0.007 | 0.15 | | 0.273 | $F_{27, 325} = 1.26$ | 0.18 |
| | | | < 0.001 | 0.002 | | | | |
| | | | 0.37 | 0.40 | | | | |

* Pillai's trace was used instead of Wilks' lambda when Box's M test was $P < 0.001$ (see text).

to the three size-paired pale and dark species (*M. christineae* vs. **M. yuma**; *M. navajo* vs. **M. kennedyi**; *M. mexicanus*-02 vs **M. mendax-03**), mean *D* (using estimated marginal means) was 1.44× larger for *M. christineae* (20.81 μm) compared to **M. yuma** (14.42 μm), 1.43× larger for *M. navajo* (20.41 μm) compared to **M. kennedyi** (14.29 μm), and 1.41× larger for *M. mexicanus*-02 (20.78 μm) compared to **M. mendax-03** (14.79 μm) (Table 1).

Mesosoma length was an important covariate in the model (Wilks' λ = 0.398, $F_{3,65}$ = 32.8, $P < 0.001$), indicating that the covariate adjusted values of the outcome. Tests of between-subjects effects differed for all three variables (eye area: $F_{1,67}$ = 88.9, $P < 0.001$; facet number: $F_{1,67}$ = 95.3, $P < 0.001$; mean *D*: $F_{1,67}$ = 12.8, $P = 0.001$). All three eye features increased with body size within all six species (Fig 5).

Diameter of the anterior ocellus differed across species after controlling for mesosoma length (ANCOVA: tests of between-subject effects; $F_{7,98}$ = 69.6, $P < 0.001$; Fig 6), but the species × mesosoma length interaction term was not significant ($F_{7,91}$ = 0.96, $P = 0.47$). Pairwise comparisons between all species pairs using LSD tests showed that anterior ocellus diameter usually was larger for dark species, though the diameter for one pale species (*M. christineae*) overlapped with this group. Ocellus diameter for the other four pale species (*M. testaceus*, *M. mexicanus*-01, *M. mexicanus*-02, *M. navajo*) was smaller than all other congeners (Fig 6). Note that *M. navajo* was not included in our statistical analysis, but it was placed lowest in this group post-hoc because the anterior ocellus was lacking in 11 of 12 workers.

Mesosoma length was an important covariate in the model based on the tests of between-subjects effects ($F_{1,98}$ = 171.2, $P < 0.001$). Diameter of the anterior ocellus increased with body size within all species except *M. navajo*. Presence of the anterior ocellus also was associated with body size in *M. mexicanus*-01 and *M. mexicanus*-02, as workers with a mesosoma length $< \approx 2.2$ mm lacked an anterior ocellus while those with a mesosoma length $> \approx 2.2$ mm

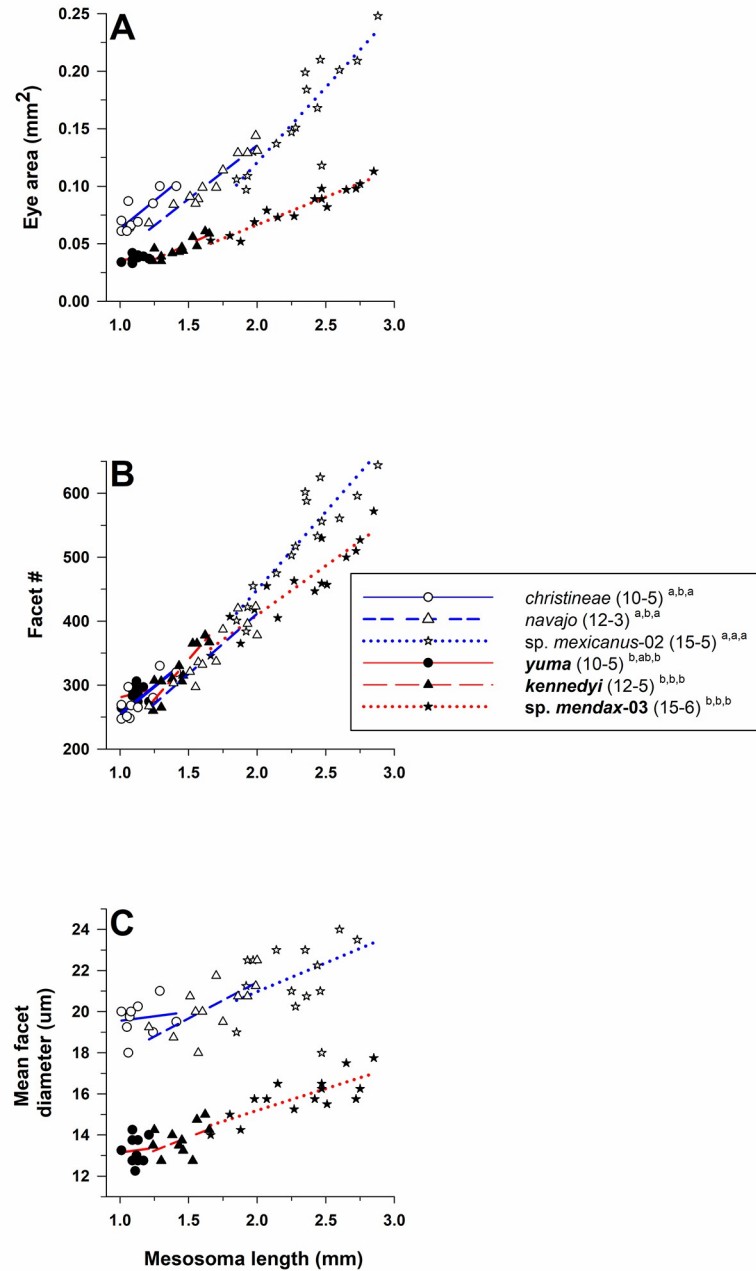

**Fig 5.** Eye area (mm$^2$) **(A)**, facet number **(B)**, and mean facet diameter ($D$) (μm) **(C)** for species of *Myrmecocystus* (subfamily Formicinae: tribe Lasiini). Three species are pale (open symbols, normal font, and blue lines: *M. christineae*, *M. navajo*, *M. mexicanus*-02), and three species are dark (filled symbols, **bold** font, red lines: **M. yuma**, **M. kennedyi**, **M. mendax-03**) (see text). For each species, number of workers examined and number of colonies they were derived from is given in parentheses. Significant differences ($P < 0.05$) among species are denoted after each species name by the letters *a*–*c*: $a > b > c$; the three sets of letters for each species correspond to panels A, B, and C, respectively. Groupings are based on univariate F tests within MANCOVA using the estimated marginal means followed by pairwise comparisons using a least significant differences test (see text).

had this ocellus, with ocellus diameter increasing with body size in these latter workers (Fig 6). Both posterior ocelli usually were present, but tiny, in workers that lacked an anterior ocellus.

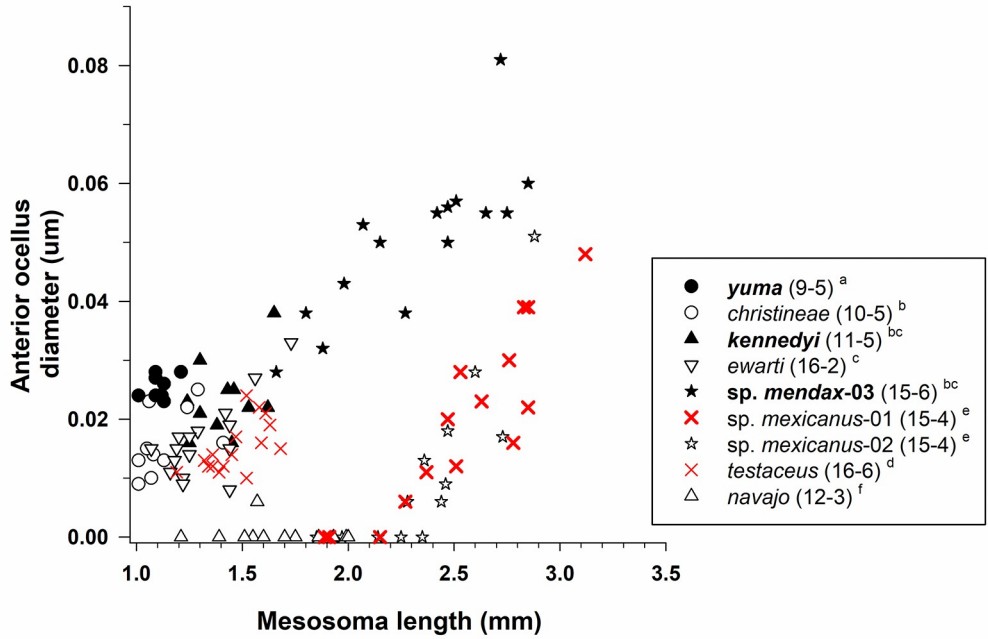

**Fig 6. Anterior ocellus diameter for species of *Myrmecocystus* (subfamily Formicinae: Tribe Lasiini).** Six species are pale (open or red symbols and normal font: *M. christineae, M. ewarti, M. navajo, M. testaceus, M. mexicanus*-01, *M. mexicanus*-02), and three species are dark (filled symbols and **bold** font: ***M. yuma, M. kennedyi, M. mendax*-03**) (see text). For each species, number of workers examined and number of colonies they were derived from is given in parentheses. Significant differences (*P* < 0.05) among species are denoted after each species name by the letters *a–f*: *a > b > c > d > e > f*; the three sets of letters for each species correspond to panels A, B, and C, respectively. Groupings are based on univariate F tests within ANCOVA followed by pairwise comparisons of the estimated marginal means using a least significant differences test (see text).

**Aphaenogaster.** Eye structure (eye area, facet number, facet diameter) differed among species of *Aphaenogaster* after controlling for mesosoma length (MANCOVA: Wilks' λ = 0.042, $F_{9,76}$ = 22.6, *P* < 0.001). Tests of between-subject effects demonstrated that all three dependent variables differed across species (eye area: $F_{3,33}$ = 125.0, *P* < 0.001; facet number: $F_{3,33}$ = 146.8, *P* < 0.001; mean *D*: $F_{3,33}$ = 31.9, *P* < 0.001; Fig 7). Based on the estimated marginal means, pairwise comparisons across all species pairs using an LSD test showed that all three eye measures were higher for the pale *A. megommata* than for all three dark congeners (*P* < 0.001). Mean *D* (using estimated marginal means) was 1.15× larger for *A. megommata* (22.78 μm) compared to ***A. occidentalis*** (19.87 μm), 1.13× larger than that for ***A. patruelis*** (20.22 μm), and 1.23× larger than that for ***A. boulderensis*** (18.49 μm) (Table 1).

Mesosoma length was an important covariate in the model (Wilks' λ = 0.502, $F_{3,31}$ = 10.2, *P* < 0.001), and tests of between-subjects effects differed for eye area ($F_{1,33}$ = 28.1, *P* < 0.001) and facet number ($F_{1,33}$ = 17.1, *P* < 0.001), but not for mean *D* ($F_{1,33}$ = 2.7, *P* = 0.11). These patterns were evidenced in that eye area and facet number increased with body size within all three species (Fig 7; ***A. boulderensis*** excluded because of small sample size), while mean *D* increased with body size for *A. megommata* and ***A. patruelis***, but it decreased with body size for ***A. occidentalis*** (Fig 7).

**Temnothorax.** Eye structure (facet number, facet diameter) differed among species of *Temnothorax* after controlling for mesosoma length (MANCOVA: Wilks' λ = 0.054, $F_{4,48}$ = 39.5, *P* < 0.001). The tests of between-subject effects demonstrated that both facet number and facet diameter differed across species (facet number: $F_{2,25}$ = 20.8, *P* < 0.001; mean *D*: $F_{2,25}$ = 53.4, *P* < 0.001; Fig 8). Based on estimated marginal means, pairwise comparisons across all

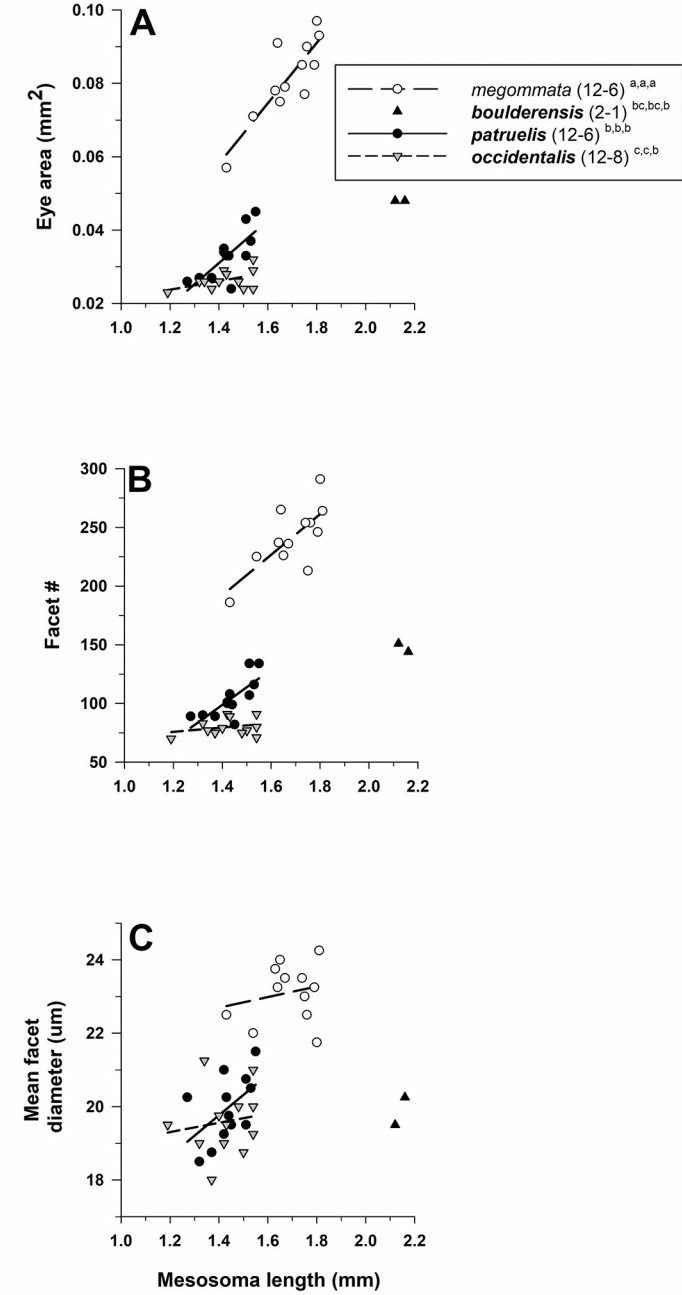

**Fig 7.** Eye area (mm$^2$) **(A)**, facet number **(B)**, and mean facet diameter ($D$) (μm) **(C)** for species of *Aphaenogaster* (subfamily Myrmicinae: tribe Stenammini). *Aphaenogaster megommata* is pale (open symbols and regular font), while **A. boulderensis**, **A. occidentalis**, and **A. patruelis** are dark (filled symbols and **bold** font) (see text). For each species, number of workers examined and number of colonies they were derived from is given in parentheses. Significant differences ($P < 0.05$) among species are denoted after each species name by the letters *a–c*: *a > b > c*; the three sets of letters for each species correspond to panels A, B, and C, respectively. Groupings are based on univariate F tests within MANCOVA using the estimated marginal means followed by pairwise comparisons using a least significant differences test (see text).

species pairs using a LSD test showed that mean $D$ was larger for the pale *T*. sp. BCA-5 than for the two dark congeners, and that facet number was higher in **T. tricarinatus** than for both **T. neomexicanus** and *T*. sp. BCA-5 (Fig 8). Relative eye area also differed across species (one-

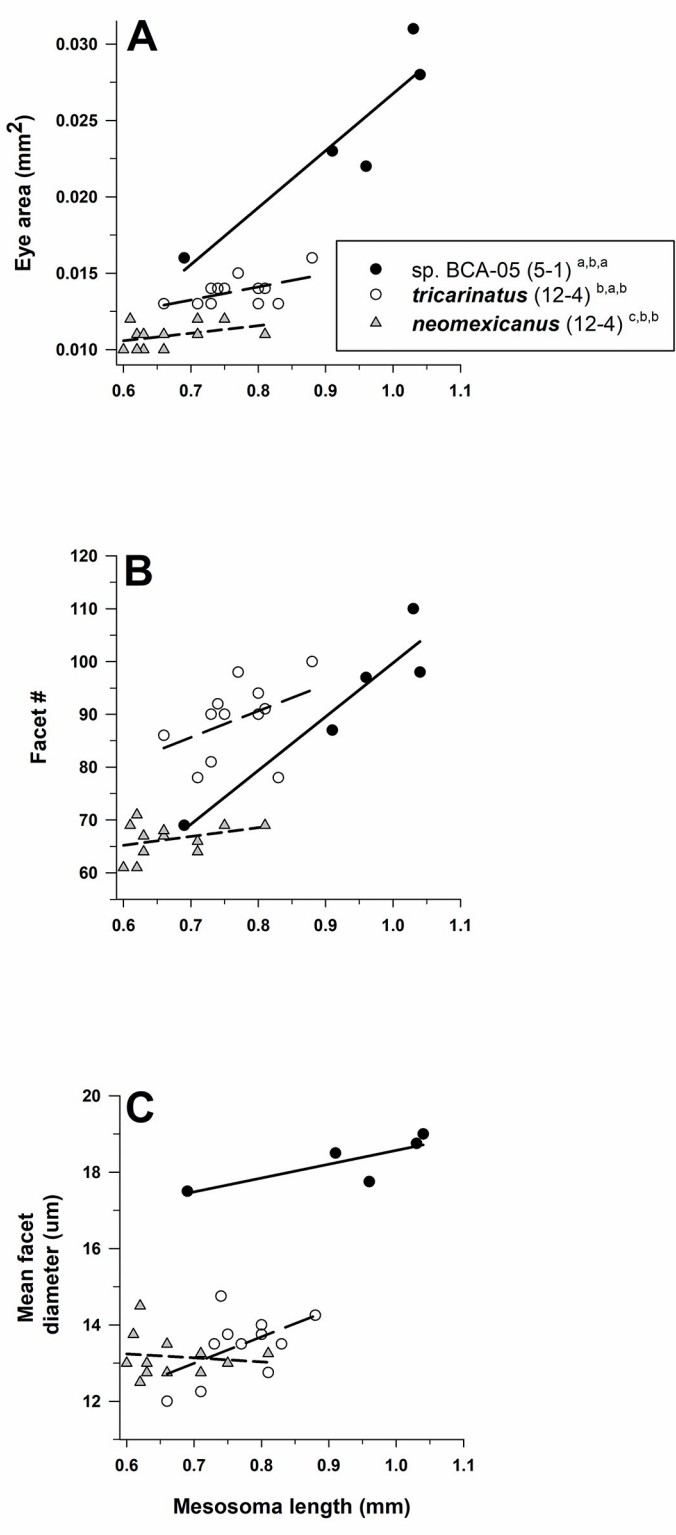

**Fig 8.** Eye area ($mm^2$) **(A)**, facet number **(B)**, and mean facet diameter ($D$) (μm) **(C)** for species of *Temnothorax* (subfamily Myrmicinae: tribe Crematogastrini). *Temnothorax* sp. BCA-5 is pale (open symbols and regular font), while *T. neomexicanus* and *T. triacarinatus* are dark (filled symbols and **bold** font) (see text). For each species, number of workers examined and number of colonies they were derived from is given in parentheses. Significant differences ($P < 0.05$) among species are denoted after each species name by the letters *a–c*: *a > b > c*; the three sets of letters for

each species correspond to panels A, B, and C, respectively. Groupings for facet number and facet diameter are based on univariate F tests within MANCOVA using the estimated marginal means followed by pairwise comparisons using a least significant differences test (see text); groupings for eye area are based on a one-way ANOVA (see text).

way ANOVA: $F_{2,26}$ = 44.0, $P < 0.001$), with size being largest for *T.* sp. BCA-05, intermediate for ***T. tricarinatus***, and smallest for ***T. neomexicanus*** (LSD test, $P < 0.05$; Fig 8). Mean *D* (using estimated marginal means) was 1.23× larger for *T.* sp. BCA-5 (17.76 µm) compared to ***T. neomexicanus*** (14.43 µm) and 1.32× larger than that for ***T. tricarinatus*** (13.42 µm) (Table 1).

Mesosoma length was an important covariate in the model (Wilks' λ = 0.546, $F_{2,24}$ = 9.97, $P < 0.001$), and tests of between-subjects effects differed for facet number ($F_{1,25}$ = 20.8, $P < 0.001$), but not for mean *D* ($F_{1,25}$ = 3.9, $P = 0.058$). Eye area and facet number increased with body size within all three species, while mean *D* increased with body size for *T.* sp. BCA-5 and ***T. tricarinatus***, but it decreased with body size for ***T. neomexicanus*** (Fig 8).

**Veromessor.** Eye structure (eye area, facet number, facet diameter) differed among species of *Veromessor* (MANCOVA: Wilks' λ = 0.024, $F_{27,351}$ = 33.9, $P < 0.001$). The tests of between-subject effects demonstrated that all three dependent variables differed across species (eye area: $F_{9,122}$ = 117.6, $P < 0.001$; facet number: $F_{9,122}$ = 124.4, $P < 0.001$; mean *D*: $F_{9,122}$ = 12.5, $P < 0.001$; Fig 9). Based on the estimated marginal means, pairwise comparisons across all species pairs using a LSD test showed that eye area was larger for the pale *V.* RAJ-*pseu* than for the dark ***V. smithi***, and eyes were larger for both species than for the pale *V. lariversi*. Eye area was smaller for all other dark congeners ($P < 0.01$; Fig 9). Facet number was higher for *V.* RAJ-*pseu* than for ***V. smithi*** and *V. lariversi*, and facet number for these three species was higher than for all other dark congeners. Mean *D* (using estimated marginal means) was highest for ***V. smithi*** and *V.* RAJ-*pseu*, followed by *V. lariversi* and *V. julianus*, with the two latter species overlapping with *V.* RAJ-*pseu* but not ***V. smithi***. Mean *D* was lower for all other dark congeners ($P < 0.01$, Fig 9; Table 1).

Mesosoma length was an important covariate in the model (Wilks' λ = 0.200, $F_{3,120}$ = 160.2, $P < 0.001$), and tests of between-subjects effects differed for all three variables (eye area: $F_{1,122}$ = 486.0, $P < 0.001$; facet number: $F_{1,122}$ = 254.2, $P < 0.001$; mean *D*: $F_{1,122}$ = 40.3, $P < 0.001$). Eye area and facet number increased with body size within all 10 species of *Veromessor*, and mean *D* increased for all species except ***V. smithi*** and ***V. chamberlini*** (Fig 9).

## Detailed eye measurements

**Variation in interommatidial angle ($\Delta\phi$).** Genus has an important effect on $\Delta\phi$ (ANCOVA: $F_{3,32}$ = 10.1, $P < 0.001$). Although the isolated effect of activity period over $\Delta\phi$ showed a weak effect ($F_{1,32}$ = 4.0, $P = 0.055$), the interaction of genus × activity period showed a strong effect ($F_{3,32}$ = 7.3, $P < 0.001$), which indicates that the four genera differed in the direction and magnitude of differences in $\Delta\phi$. Across all species, values of $\Delta\phi$ ranged from 3.5–7° (Fig 10).

Pale species had larger mean $\Delta\phi$'s in *Myrmecocystus* (t-test: $t_8$ = -3.4, $P = 0.01$) and *Veromessor* ($t_6$ = -0.3, $P = 0.77$), but dark species had larger $\Delta\phi$'s in *Aphaenogaster* ($t_8$ = -3.6, $P = 0.007$) and *Temnothorax* ($t_8$ = 2.7, $P = 0.027$) (Fig 10). Across genera, $\Delta\phi$ was lowest for *Myrmecocystus* and *Veromessor* and highest for *Aphaenogaster* and *Temnothorax* (Tukey's HSD test, $P < 0.05$) (Fig 10).

Mesosoma length was an important covariate in the above model ($F_{1,31}$ = 5.4, $P = 0.026$), a fact largely influenced by $\Delta\phi$ decreasing in larger workers of *Temnothorax* and *Veromessor*,

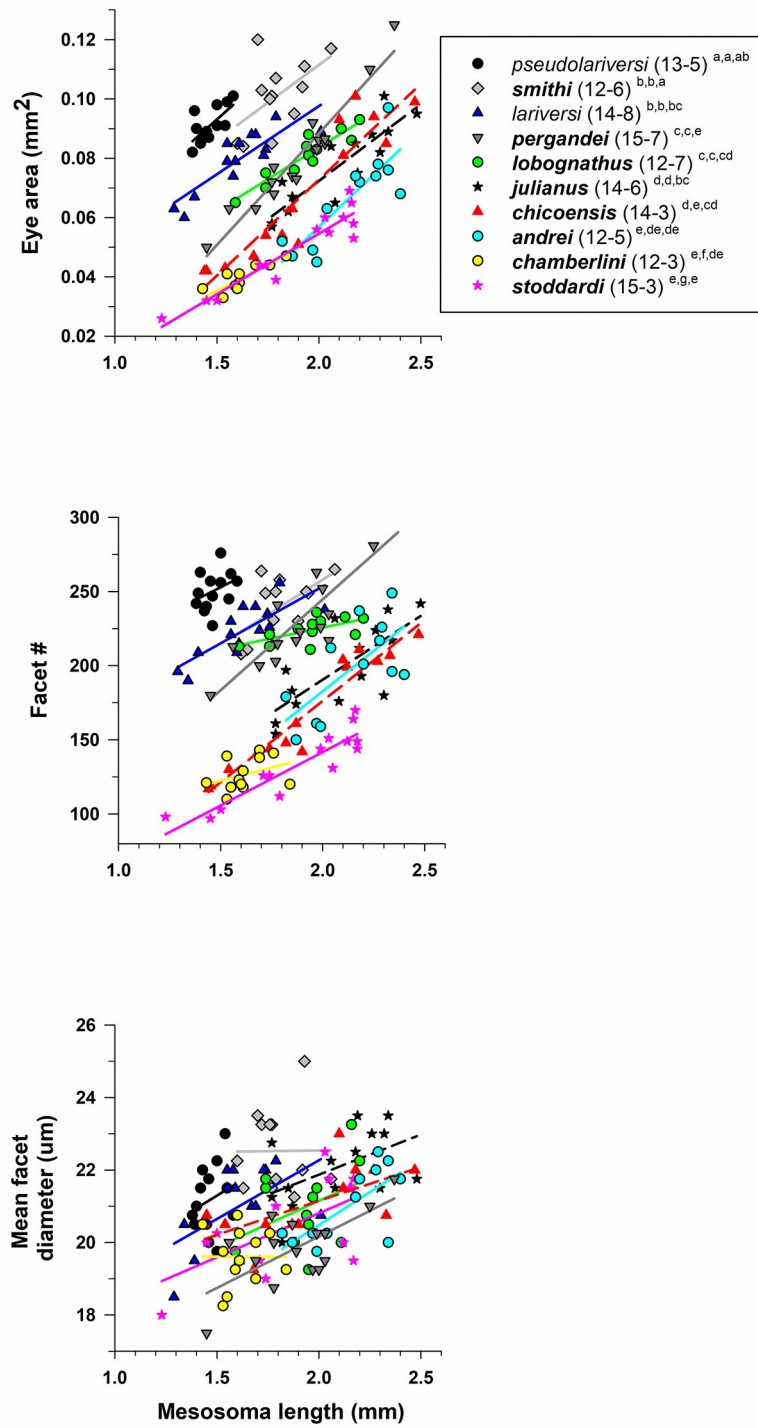

**Fig 9.** Eye area (mm$^2$) **(A)**, facet number **(B)**, and mean facet diameter (*D*) (µm) **(C)** for species of *Veromessor* (subfamily Myrmicinae: tribe Stenammini). *Veromessor lariversi* and *V*. RAJ-*pseu* are pale (open symbols and regular font), while the other eight species are dark (filled symbols and **bold** font) (see text). For each species, number of workers examined and number of colonies they derived from is given in parentheses. Significant differences ($P < 0.01$) among species are denoted after each species name by the letters *a–g*: $a > b > c > d > e > f > g$; the three sets of letters for each species correspond to panels A, B, and C, respectively. Groupings are based on univariate F tests within MANCOVA using the estimated marginal means followed by pairwise comparisons using a least significant differences test (see text).

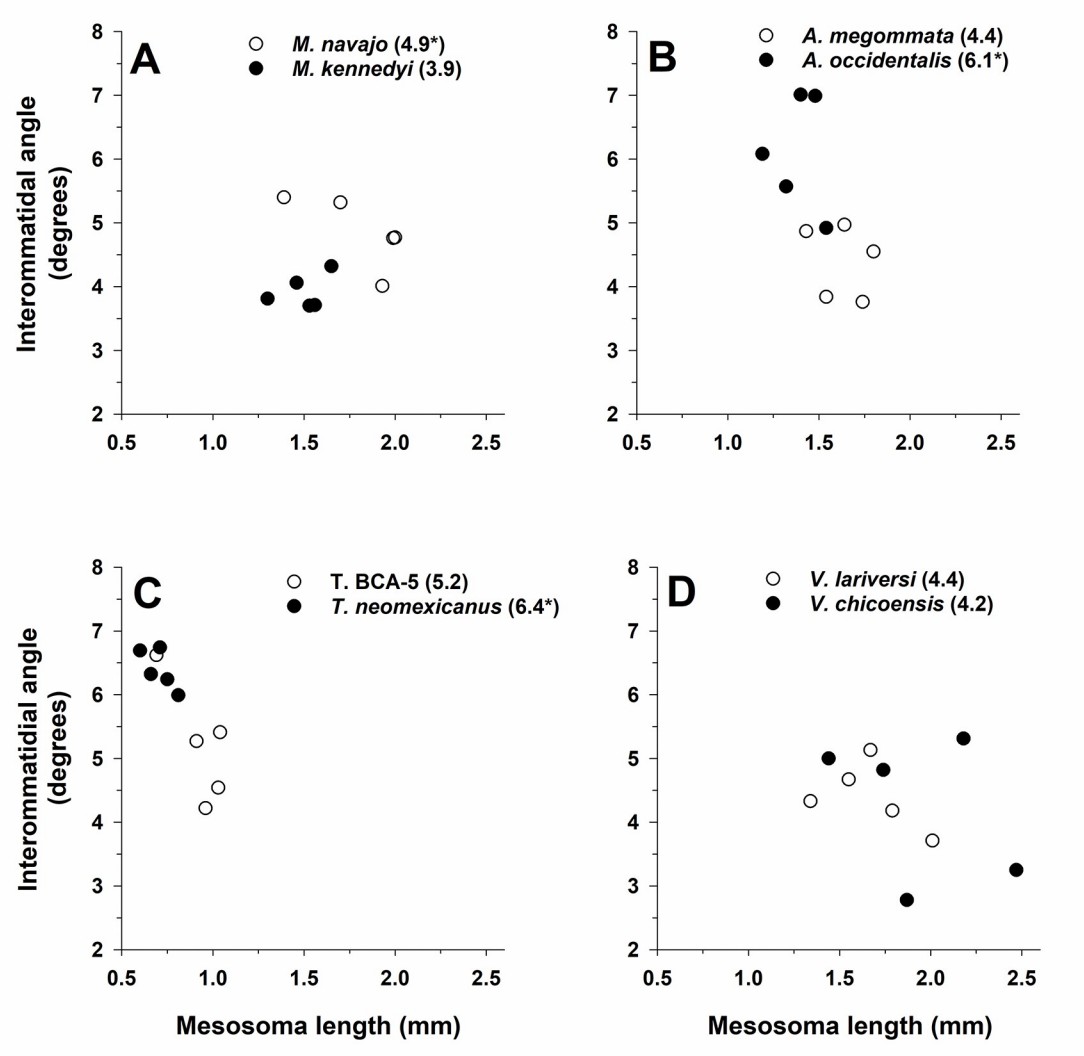

**Fig 10.** Interommatidial angle ($\Delta\phi$) for one pale (open circles and regular font) and one dark (filled circles and bold font) (see text) species in each of four ant genera: (**A**) *Myrmecocystus*, (**B**) *Aphaenogaster*, (**C**) *Temnothorax*, and (**D**) *Veromessor*. All plots have the same x-axis and y-axis scaling in order to visualize differences between light and dark species across genera. Mean $\Delta\phi$ (in degrees) is given after each species name with an asterisk denoting the species with a significant larger $\Delta\phi$ based on a t-test ($P < 0.05$). The significant interaction of genus × activity period is shown by larger $\Delta\phi$'s for pale species of *Myrmecocystus* and *Veromessor*, whereas $\Delta\phi$ was larger for dark species of *Aphaenogaster* and *Temnothorax*. Sample size is $n = 5$ for each species.

and not varying with mesosoma length within species of *Myrmecocystus* and *Aphaenogaster* (Fig 10).

**Eye parameter ($\rho$).** The ANCOVA for $\rho$ was significant for genus ($F_{3,32} = 11.6$, $P < 0.001$), activity period ($F_{1,32} = 11.2$, $P = 0.002$), and the interaction of genus × activity period ($F_{3,32} = 13.1$, $P < 0.001$). As expected, overall $\rho$ was higher for pale (mean = 1.70) than for dark species (mean = 1.51), however, a significant genus × activity period interaction indicated differences in direction and magnitude of these differences (Fig 11). Pale species had the larger mean $\rho$ in *Myrmecocystus* (t-test: $t_{8\ df} = -8.9$, $P < 0.001$), while values did not differ between pale and dark species in the other three genera (*Veromessor*: $t_8 = -0.3$, $P = 0.74$; *Temnothorax*: $t_4 = -1.7$, $P = 0.16$; *Aphaenogaster*: $t_8 = -2.2$, $P = 0.058$). Across genera $\rho$ was highest

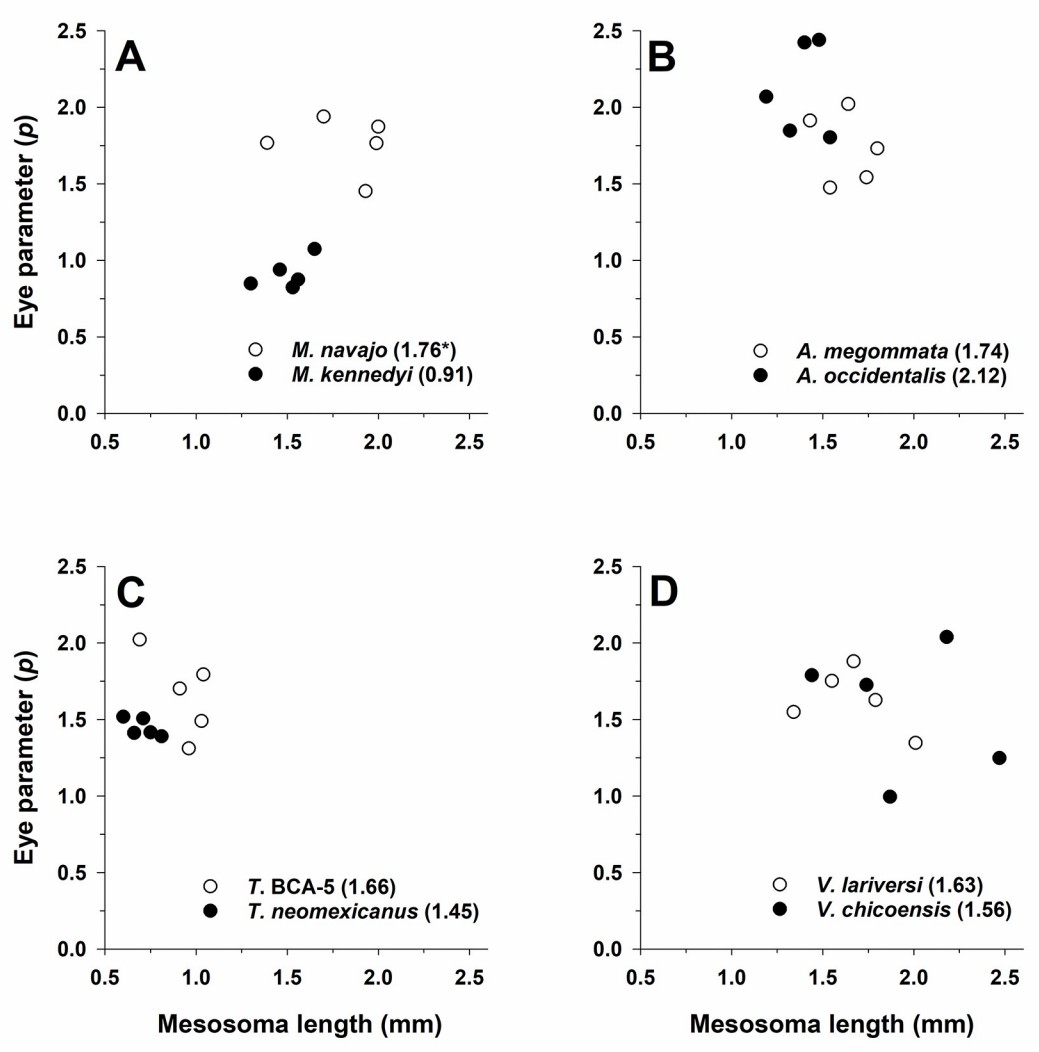

**Fig 11.** Eye parameter ($\rho$) for one pale (open circles and regular font) and one dark (filled circles and bold font) (see text) species in each of four ant genera: (**A**) *Myrmecocystus*, (**B**) *Aphaenogaster*, (**C**) *Temnothorax*, and (**D**) *Veromessor*. All plots have the same x-axis and y-axis scaling in order to visualize differences between light and dark species across genera. Mean *p* is given after each species name with an asterisk denoting the species with a significant larger *p* based on a t-test ($P < 0.05$). The significant interaction of genus × activity period is shown by larger differences between light-colored and dark-colored species of *Aphaenogaster* compared to those in the other three genera. Sample size is $n = 5$ for each species.

for *Aphaenogaster*, intermediate for *Temnothorax* and *Veromessor*, and lowest for *Myrmecocystus* (Tukey's HSD test, $P < 0.05$; Fig 11).

Mesosoma length was not an important covariate in the model ($F_{1,31} = 1.6$, $P > 0.20$). The $\rho$ was not positively or negatively correlated with body size for any of the examined species (Fig 11).

**Visual field span.** The ANCOVA for visual field span was significant for genus ($F_{3,32} = 53.6$, $P < 0.001$), activity period ($F_{1,32} = 151.7$, $P < 0.001$), and the interaction of genus × activity period ($F_{3,32} = 14.8$, $P < 0.001$). Visual field span was greater for pale (mean = 98.8°) than for dark species (mean = 73.0°), and the significant genus × activity period interaction indicated that differences between the visual field of pale and dark species were larger in some genera, e.g., *Aphaenogaster*, than others (Fig 12). Though not always significantly different, pale species had a larger mean visual field in all four genera (*Myrmecocystus* t-

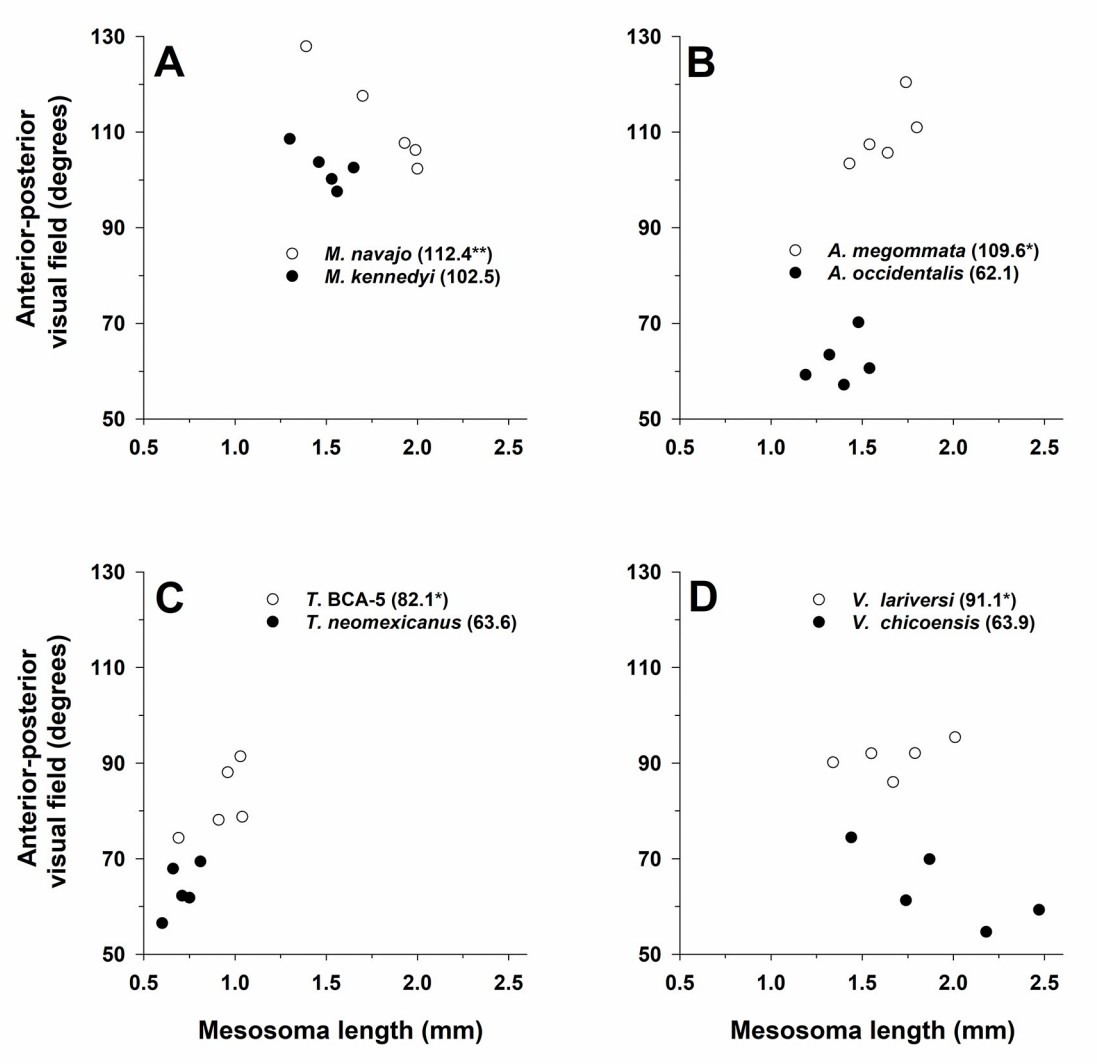

**Fig 12.** Anterior-posterior visual field span (in degrees) for one pale (open circles and regular font) and one dark (filled circles and bold font) (see text) species in each of four ant genera: (**A**) *Myrmecocystus*, (**B**) *Aphaenogaster*, (**C**) *Temnothorax*, and (**D**) *Veromessor*. All plots have the same x-axis and y-axis scaling in order to visualize differences between pale and dark species across genera. Mean visual field span (in degrees) is given after each species name with an asterisk denoting the species with a significant larger visual field based on a t-test ($P < 0.05$); the double asterisk denotes that the t-test was not significant, but that the visual field was significantly larger when including mesosoma length as a covariate. The significant interaction of genus × activity period is shown by larger differences between pale and dark species of *Aphaenogaster* compared to those in the other three genera. Sample size is $n = 5$ for each species.

test $t_{5\ df} = -1.90$, $P = 0.10$; *Aphaenogaster*: $t_8 = 12.7$, $P < 0.001$; *Temnothorax*: t-test $t_8 = -4.6$, $P = 0.002$; *Veromessor*: $t_7 = -6.9$, $P < 0.001$). Across genera the visual field was greatest for *Myrmecocystus*, intermediate for *Aphaenogaster*, and smallest in *Temnothorax* and *Veromessor* (Tukey's HSD test, $P < 0.05$; Fig 12).

Mesosoma length was not an important covariate in the above model ($F_{1,31} = 2.6$, $P = 0.11$), in part, because of the differing patterns exhibited across species. For example, visual field span was positively correlated with mesosoma length in *A. megommata*, **A. occidentalis**, *T.* BCA-5, **T. neomexicanus**, and *V. lariversi*, but these two variables were negatively correlated in **M. kennedyi**, *M. navajo*, and **V. chicoensis** (Fig 12).

**Table 4. Repeated measures ANOVA results for regional variation in facet diameter for one pale (normal font) and one dark species (bold font) (see text) of *Myrmecocystus* and *Veromessor* (*n* = 12 per species for *Myrmecocystus*; *n* = 14 per species for *Veromessor*).**

| Species | Multivariate test (Wilks' lamba) | Mauchly's test of sphericity | | | Within-subjects effects |
| --- | --- | --- | --- | --- | --- |
| | | Mauchly's W | Chi-square, df | P | F, df, P |
| *M. navajo* | 0.273, $P = 0.022$ | 0.427 | 8.02, 9 df | 0.539 | 7.04, 4 df, $P < 0.001$ |
| **M. kennedyi** | 0.671, $P = 0.469$ | NA* | NA | NA | NA |
| *V. lariversi* | 0.117, $P < 0.001$ | 0.302 | 13.67, 9 df | 0.139 | 19.60, 4 df, $P < 0.001$ |
| **V. chicoensis** | 0.090, $P < 0.001$ | 0.269 | 14.98, 9 df | 0.095 | 13.69, 4 df, $P < 0.001$ |

* NA (= not available) given that the Wilks' lambda value was not significant.

The maximum visual span for pale species was 128° for *M. navajo* and 121° for *A. megommata*; the maximum for all other species was < 120°. Because their eyes are located on the side of the head, the center of these relatively small visual field spans was directed laterally. This means that, in the ants examined here, there was no forward part of the visual field for either eye directed toward the mouthparts, and so there was no anterior region of binocular vision.

**Regional variation in *D*.** *D* varied regionally in three (*M. navajo*, ***V. chicoensis***, *V. lariversi*) of the four species; all three species met the assumption of sphericity (Table 4; Fig 13). Based on a post-hoc LSD test, the anterior and ventral facets were largest in *M. navajo*, ventral facets were largest in ***V. chicoensis***, and lateral and ventral facets were largest in *V. lariversi*. *D* did not vary across regions in ***M. kennedyi***, but the ventral facets had the largest mean *D* (Table 4; Fig 13).

**Additional pale ant species with enlarged eyes.** We found numerous additional pale ant species with enlarged eyes on Antweb (Table 5 and S1 Table). The combination of pale color and enlarged eyes occurred in numerous additional species of *Temnothorax* from both the Old and New World, as well as in two additional genera–*Dorymyrmex* and *Iridomyrmex* (subfamily Dolichoderinae), which comprises a third subfamily containing pale species. In *Temnothorax*, this combination of traits evolved in at least two species groups in the United States and Mexico (*T. silvestrii* and *T. tricarinatus*) and in at least one species group in northern Africa (*T. laurae*) (Table 5). Interestingly, some pale species of *Temnothorax* did not have enlarged eyes, e.g., *T. agavicola T. colkendolpheri*, and *T. indra*.

## Discussion

### Foraging and cuticular color

As predicted, worker color was correlated with foraging time across all four genera of ants. Specifically, pale coloration is linked to nocturnal foraging in these and potentially other pale ants. Alternatively, dark species usually forage diurnally, but some species also forage nocturnally during warm seasons, and several species are largely matinal-crepuscular-nocturnal foragers. The above two patterns support the hypothesis that activity pattern can produce strong selection pressure on body coloration and eye structure. These relationships are found in a number of taxa of ants and other organisms [15, 16, 26], but it is not a necessary phenotype given the numerous taxa living in similarly dim light conditions that have retained their pigmentation.

A species-level phylogeny is available for all four genera such that we can infer the direction of trait evolution. These phylogenies indicate that pale color is a derived trait in *Aphaenogaster* [46], *Temnothorax* [49], and *Veromessor* (M. Borowiec & R.A. Johnson, unpub. data), i.e., all most recent common ancestors of pale species were dark, but that it is an ancestral trait in *Myrmecocystus* [45], i.e., pale color was a basal trait in this genus and that these species gave

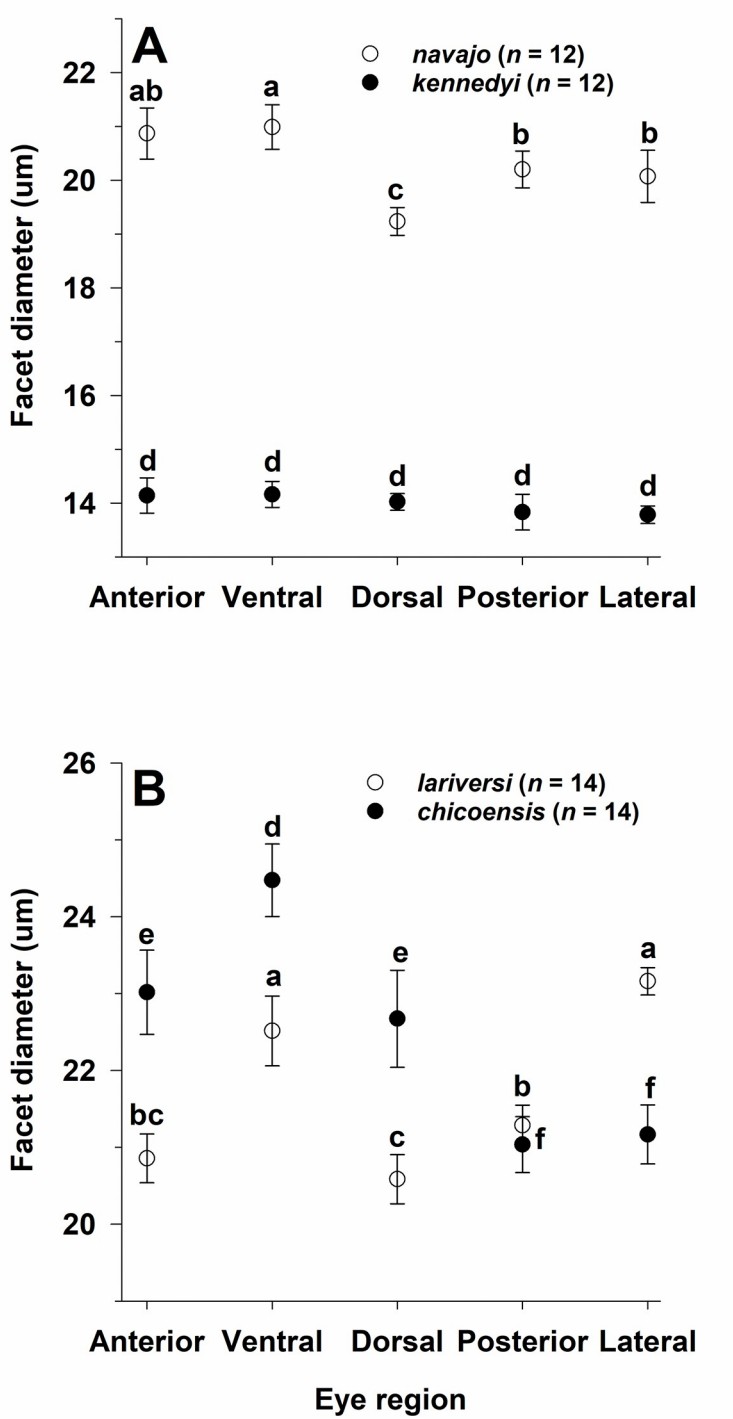

**Fig 13.** Regional variation in facet diameter for one pale (open circles and regular font) and one dark (filled circles and bold font) (see text) species in each of two ant genera: **(A)** *Myrmecocystus* and **(B)** *Veromessor*. Significant differences within each species are denoted by the letters *a–c*: *a > b > c* for pale species; *d–f*: *d > e > f* for dark species. Groupings for each species are based on a repeated-measures ANOVA followed by a least significant differences test. Sample size is given after each species name.

**Table 5. Additional pale ant species (see text) with enlarged eyes based on photographs examined on Antweb (www.antweb.org).** Species are listed alphabetically by subfamily, genus, species group, species, and subspecies. High resolution photographs of each species can be viewed by going to https://www.antweb.org/advSearch.do, then placing the genus and species name in the advanced search box. Brightness values are given as mean (*n*) (see text). Relative eye size was calculated as eye area/mesosoma length.

| Species | Brightness value | Type locality (country) | Relative eye size |
|---|---|---|---|
| **Subfamily Dolichoderinae–genus *Dorymyrmex*** | | | |
| *D. ensifer* Forel | 72.5 (2) | ARGENTINA | 0.0331 |
| *D. ensifer laevigatus* Gallardo | 72.7 (1) | ARGENTINA | 0.0323 |
| *D. ensifer weiseri* Santschi | 72.0 (1) | ARGENTINA | 0.0340 |
| *D. exsanguinus* Forel | 71.3 (3) | ARGENTINA | 0.0370 |
| *D exsanguinus anaemicus* Santschi | 76.3 (1) | ARGENTINA | 0.0384 |
| *D. morenoi patagon* Santschi | 85.3 (1) | ARGENTINA | 0.0344 |
| *D.* nr. *Morenoi* | 72.3 (1) | ARGENTINA | 0.0383 |
| Dark congeners[+] | | | 0.0249 |
| **Subfamily Dolichoderinae–genus *Iridomyrmex*** | | | |
| *I. macrops* Heterick & Shattuck | 73.7 (1) | AUSTRALIA | 0.0341 |
| **Dark congeners** | | | 0.0278 |
| **Subfamily Myrmicinae–genus *Temnothorax*** | | | |
| *T. laurae*-group[*] | | | |
| *T. arenarius* (Santschi) | 81.2 (2) | TUNISIA | 0.0383 |
| *T. arenarius fusciventris* (Santschi) | 78.7 (1) | TUNISIA | 0.0353 |
| *T. canescens* (Santschi) | 73.3 (1) | SPAIN | 0.0246 |
| *T. laciniatus* (Stitz) | 75.3 (1) | ALGERIA | 0.0288 |
| *T. laurae* (Emery) | 72.7 (1) | TUNISIA | 0.0273 |
| *T. laurae rosae* (Santschi) | 77.0 (1) | TUNISIA | 0.0314 |
| *T. megalops* (Hamman & Klemm) | 72.3 (1) | SUDAN | 0.0267 |
| *T. mpala* Prebus | 73.0 (1) | KENYA | 0.0261 |
| *T. naeviventris* (Santschi) | 73.4 (3) | TUNISIA | 0.0330 |
| *T. naeviventris kefensis* (Santschi) | 74.0 (1) | TUNISIA | 0.0243 |
| *T. tricarinatus*-group[*] | | | |
| *T. bestelmeyeri* (MacKay) | 75.0 (1) | USA | 0.0303 |
| *T. coleenae* (MacKay) | 78.0 (1) | USA | 0.0286 |
| *T. liebi* (MacKay) | 83.3 (1) | USA | 0.0291 |
| **Dark congeners** | | | 0.0224 |

[+] The last line for each genus lists the largest relative eye size for the five dark congeners (see S1 Table).

[*] Species groups for *Temnothorax* as per Matt Prebus (pers. comm.).

rise to dark congeners. Van Elst [45] also determined that the subgenus *Myrmecocystus* (all pale species) and the genus *Myrmecocytus* as a whole most likely originated from a nocturnal ancestor. Interestingly, the sister genus, *Lasius* [45], contains numerous pale species that are largely subterranean with very small eyes [72, 73].

## Compound eye morphology

We hypothesized that the eyes of pale ant species maximized sensitivity at the expense of resolution, which is the pattern typical for most insects with apposition eyes [35, 74]. The predicted features to enhance sensitivity include larger eye size, larger facets, larger $\Delta\phi$'s, and a higher $\rho$. Using pale body color as an indicator of nocturnal activity in ants demonstrated consistent correlated adaptations in eye structure across four genera of ants in two subfamilies. When

controlled for body size, pale species exhibited convergent morphology for some characters, but not for others: all pale species (except *V. lariversi*) had larger eyes, a larger *D*, and a larger visual span compared to their dark congeners. *Aphaenogaster megommata* and *V.* RAJ-*pseu* also possessed more eye facets than their dark congeners. Alternatively, $\Delta\phi$ and $\rho$ displayed variable patterns both within and among genera.

Sensitivity (light gathering potential) of an eye is a function of four variables that effect photon capture–*D*, rhabdom diameter, rhabdom length, and focal length [see 29]. Facet area [$\pi/4 \times D^2$] is a particularly important indicator in the differences in light capturing ability of the eyes between pale and dark species in each genus [35]. This difference was highest for *Myrmecocystus* with facet area for pale species about 2.0–2.1-fold higher than for their paired dark species (calculated using estimated marginal means as [$\pi/4 \times D^2_{pale}$]/ [$\pi/4 \times D^2_{dark}$]; see Table 1), about 1.3–1.5-fold higher for *A. megommata* compared to its dark congeners, and about 1.5–1.7-fold higher for *T.* sp. BCA-5 compared to its dark congeners. Alternatively, for *Veromessor*, facet area was highest in the dark **V. smithi**. The mean difference was about 1.05–1.10× higher for **V. smithi** than for *V. lariversi* and *V.* RAJ-*pseu*; all other dark congeners had a smaller *D*. Moreover, our measurements of *D* documented clear differences in eye structure and sensitivity between pale and dark species. Future studies should examine the other three above variables so as to more completely understand the suite of structural adaptations possessed by eyes of these pale species.

Although not examined, the eyes in pale species of *Myrmecocytus* were more protruding and dome-shaped compared to the more flattened eyes of their dark congeners (Fig 1). These more bulging, dome-shaped eyes could result in a greater radius of curvature and possibly a greater visual span field, as well as space for more facets within a given eye area.

Our results show variation in the strength of the relationship between activity patterns, body color, and eye structures that suggest other variables, yet to be determined, have come into play during evolution. Interestingly, the two pale sister species of *Veromessor*, *V.* RAJ-*pseu* and *V. lariversi*, displayed different patterns of eye structure, with *V.* RAJ-*pseu* having larger eyes and more facets than *V. lariversi* (Fig 9). Additionally, eyes of the dark species **V. smithi** were smaller with fewer facets than *V.* RAJ-*pseu*, but they were larger with larger *D*'s compared to *V. lariversi*. This may result from the fact that **V. smithi** is a nocturnally-active dark species. One difference between *V. lariversi* and *V.* RAJ-*pseu* and other pale species examined herein is their more yellowish-amber to yellowish-orange color and lower B value, indicating that they are less pale than pale species in the other three genera (see Table 1; Figs 1–4). Differences in eye structure across genera along with variation across species of *Veromessor*, (e.g., *V. lariversi*, *V.* RAJ-*pseu*, and **V. smithi**) are similar to the wide variation in degree of pigment loss and eye degeneration found among cave-dwelling species that is caused by differences in divergence time and intensity of selection [18]. Similar variation occurs for nocturnal foraging bees in which many species have a relatively pale body color, and many but not all species have enlarged compound eyes and ocelli [20].

Interommatidial angles ($\Delta\phi$'s) determine the distance at which objects can be resolved, with resolution distance increasing as $\Delta\phi$ decreases. All of our species had relatively large $\Delta\phi$'s that ranged from 3.5–7$^o$, especially when compared to many common flying insects that have $\Delta\phi$'s that range from 1–3$^o$ [37]. Pale and dark species varied in their patterns of $\Delta\phi$ which were significantly larger in the dark *Temnothorax* and *Aphaenogaster*, which had small eyes with fewer facets, but was larger for the pale *Myrmecocystus* which had numerous eye facets. Moreover, $\Delta\phi$ did not decrease for pale species indicating that daily activity patterns have had little effect on the evolution of resolving power.

The eye parameter ($\rho$) measures the tradeoff between sensitivity and resolution. Insects active during high light conditions usually have low $\rho$ values that enhance resolution, whereas

species active in low light have higher $\rho$ values that often exceed 2 μm rad [36]. Across our four genera, $\rho$ was significantly higher only for the pale *M. navajo* compared to the dark **M. kennedyi**, with $\rho$ for the former species approaching 2 (Fig 11). The higher $\rho$ value for *M. navajo* resulted from the combination of larger facets and a larger $\Delta\phi$ (Fig 11). In contrast, **M. kennedyi** was the only strictly diurnal forager among all dark species (Table 1), and correspondingly it had the lowest mean $\rho$ value (0.91) among all species (Fig 11). The $\rho$ value was similar for the other three pairs of congeners, with the dark **A. occidentalis** having the highest $\rho$ value (2.12) of all species (Fig 11). The lack of consistent patterns across species of *Aphaenogaster*, *Temnothorax* and *Veromessor* likely reflects the wide range of light conditions under which dark species forage including nocturnal foraging in some seasons (Table 1).

The visual field was larger for pale species in all four genera (Fig 12). Moreover, there was no indication that these species had binocular vision in the anterior-posterior direction, that is, they cannot use their eyes for binocular depth perception. This infers that these ants do not use vision to find or capture food items, which aligns with diets that include stationary objects such as seeds, dead insects, and extrafloral nectaries. Instead, it seems likely their eyes are used for detection and orientation relative to land-based and celestial cues used in navigation (see below). In addition, our finding that visual field usually correlated with body size (positively or negatively, depending on the species), contrasted with the pattern for *Cataglyphis bicolor*, in which visual field was independent of body size [75].

Regional variation in *D* is common in insects [76], with these size differences probably related to the different selection pressures on eye structure in each region. Larger facets imply that insects have better vision from regions containing larger facets. In this study, ventral facets were significantly larger in three of the four examined species (along with anterior facets in two species) (Fig 13). This general pattern suggests that ventral facets are important for vision in both diurnal and nocturnal activity, perhaps as a mechanism for optic flow to measure distance [see 77, 78].

**Other pale ants with enlarged eyes.** Our survey of images on Antweb revealed the occurrence of pale ants with enlarged eyes compared to their dark congeners not just in the clades studied here but also in other genera. Moreover, these coupled traits appear to have evolved independently multiple times across at least three subfamilies. Given the limited scope of museum specimens available, it is likely that many pale species remain to be discovered. As these are located, our technique for measuring brightness provides a tool for mapping patterns of pigment loss within and across ant genera.

One commonality among pale species examined herein and in Table 5 is that many of these species are largely restricted to desert and semi-arid habitats. As such, these species possess visual adaptations to be nocturnal specialists in extreme environments in a manner similar to heat tolerance adaptations possessed by their thermophilic diurnal counterparts such as *Myrmecocytus kennedyi* and *Forelius* spp. [30; R.A. Johnson, pers. obs.] in the New World, and *Cataglyphis* spp. and *Melophorus bagoti* in the Old World [79, 80]. The open, exposed nature of their foraging environment lacks overstory which suggests that these species can obtain navigation cues from local landmarks via their enlarged eye facets, but probably only horizon and lunar night sky cues. However, at this point, nothing is known about navigation in any pale species, and among dark species, orientation and navigation have been examined only in the column-foraging and mostly diurnal **V. pergandei** [81–83]. There is much to learn about how ants use their eyes both at night and during the day.

**Ocelli.** Size of the anterior ocellus varied among pale and dark species of *Mymecocystus*. In larger species, the anterior ocellus was smaller in pale compared to dark species, but this difference largely disappeared for smaller species (Fig 6). The two largest pale species (*M. mexicanus*-01, *M. mexicanus*-02) also displayed size-dependent presence of the anterior ocellus as it

was present only in larger workers. The anterior ocellus also was absent in nearly all workers of the intermediate sized *M. navajo*. The pattern was mixed for smaller species because the anterior ocellus was largest for the dark **M. yuma**, intermediate for the dark **M. kennedyi** and pale *M. ewarti*, and smallest for the pale *M. testaceus* (Fig 6). These results contrast with other social insects in which the anterior ocellus is typically larger in nocturnal compared to crepuscular and diurnal flying species, e.g., bees and ants [33, 84–87] and pedestrian workers in the ant genus *Myrmecia* [88].

Absence of the anterior ocellus in workers of some pale species displays a phylogenetic component. Pale species in which some to most workers lacked the anterior ocellus fell into one clade, while all other species that always have an anterior ocellus were in two other clades [see 45]. We were unable to examine the two other pale species (*M. pyramicus*, *M. melanoticus*) because specimens were unavailable, but this phylogenetic association predicts that the anterior ocellus always is present in *M. pyramicus* and that it is only present in larger workers of *M. melanoticus* (see Fig 6). To our knowledge, these are the only known ant species in which workers display intraspecific variation in presence-absence of the anterior ocellus, making them excellent candidates to examine evolution, development, and function of the anterior ocellus, as well as how such variation affects forager orientation and navigation (see below). Foraging behavior is poorly documented in pale species, but it appears that both small and large workers of *M. mexicanus*-02, i.e., those with and without an anterior ocellus, leave the nest to forage (J. Conway, pers. comm.).

The function of ocelli in ant workers is poorly understood because most species lack ocelli (notable exceptions include the genera *Cataglyphis*, *Formica*, *Myrmecocystus*, *Polyergus* in the subfamily Formicinae; *Myrmecia* in the subfamily Myrmeciinae; *Pseudomyrmex* in the subfamily Pseudomyrmecinae). Flying insects have three ocelli that serve the general purpose of sensing polarized light for navigation and maintaining flight stability, whereas worker ants that have been studied use their ocelli to detect polarized light for navigation in *Cataglyphis bicolor* [89] or to gather light in *Myrmecia* [88]. Ocelli almost always are present in the flying queen and male ants, such that species in which the pedestrian workers possess ocelli provide an avenue to compare ocellar structure and function relative to mode of locomotion and caste. The only such comparison using the nocturnal *Myrmecia nigriceps* found structural differences between workers and males that included presence of an ocellar tapetum in males but not in workers [88].

Lastly, compound eyes and ocelli provide separate and functionally different visual pathways, so it is instructive to test for convergence in the two pathways. Two studies compared compound eyes and ocelli between nocturnal and diurnal ant species. In leafcutter ants (genus *Atta*), both the ocelli and eye facets were larger in nocturnal compared to diurnal species of both flying queens and males, while eye area was similar for species in both activity groups [85]. The other study examined workers of four species of *Myrmecia* also finding that both the ocelli and eye facets were larger in nocturnal compared to diurnal congeners, while number of facets was similar for species in both activity groups [29, 88]. Alternatively, we report here that pale species of *Myrmecocystus* had an anterior ocellus that was similar in size to smaller than comparable dark congeners, but that eye facet diameter and eye size were larger for pale compared to dark species. However, like the pattern found in *Myrmecia* [88], facet number was similar for species of *Myrmecocystus* in both activity groups.

## Conclusions

This study provides a first overview of variation in external eye structure across several ant genera that compares closely related pale and dark congeners. Our observations on body coloration and eye structure allow several statements about their visual ecology. First, the

correlation between ant body color and activity period parallels that found in other animals. The specific selective factors shaping this correlation await more detailed work on the costs and benefits of cuticular pigmentation. Second, pale, above ground foraging ants have enlarged rather than reduced or no eyes relative to their dark congeners, suggesting that vision is important for both nocturnal and diurnal species across several lineages. That pale species possess optical adaptations to maximize sensitivity, which is the pattern typical for most nocturnal insects with apposition eyes [35, 74], also suggests that vision plays a role in navigation for these nocturnal ants. Third, the visual field span and mild regional variation in *D* suggest that their eyes are not adapted to gather detailed visual information from any specific region in the space around the ant, but rather they are gathering relatively low quality information from a large part of the space around them. Fourth, the mild differences in eye structure between pale and dark species suggest both groups use their eyes in similar ways, and they are consistent with observations that these ants use their vision in navigation guided by large celestial and landmark cues. Field studies that detail foraging behavior and navigational skills would complement these data. Additional research should be done to more thoroughly determine optical sensitivity. This study only examined facet diameter, but data are needed on rhabdom diameter, rhabdom length, focal length, and neural adaptations to more completely determine and compare optical sensitivity [see 29]. We also note that activity period is the primary difference between our pale and dark species given that life history and behavior are similar for species within each genus, i.e., most species have solitary foragers that harvest seeds, or scavenge for debris, dead insects, and plant exudates [30, 50: R.A. Johnson, pers. obs.], probably using olfactory and/or tactile cues.

Of the genera examined herein, we believe *Myrmecocystus* has the most potential for further study given the consistently large variation in eye structure between pale and dark species (eye area, *D*, $\rho$, visual span), combined with the fact that most species are strongly polymorphic such that traits can be compared allometrically [see 75]. Additionally, this is the only known genus with pale species that possess ocelli, such that it provides an excellent group to examine internal eye structure and to compare evolution of both eyes and ocelli. The flying queens and males might also be examined for comparative study of the sexual castes, especially given that the queen of *M. navajo* has extremely large ocelli.

## Supporting information

**S1 Table. Brightness values and relative eye size for dark congeners from the ant genera *Dorymyrmex*, *Iridomyrmex*, and *Temnothorax*.** Phylogenies were not available for these genera, so we analyzed the first five species/nominal subspecies from each genus on Antweb (www.antweb.org). Species are listed alphabetically by subfamily, genus, species group, species, and subspecies. High resolution photographs of each species can be viewed at https://www.antweb.org/advSearch.do, then typing in the genus and species name in the advanced search box. Brightness values are given as mean (*n*) (see text). Relative eye size was calculated as eye area/mesosoma length.
(DOCX)

## Acknowledgments

We thank Brian Fisher, Michele Esposito, and Antweb for the high resolution photographs, Christian Rabeling for use of his microscope and photographic programs, and Matt Prebus for the loan of specimens. We also thank two anonymous reviewers for comments that improved this manuscript.

## Author Contributions

**Conceptualization:** Robert A. Johnson.

**Data curation:** Robert A. Johnson, Ronald L. Rutowski.

**Formal analysis:** Robert A. Johnson, Ronald L. Rutowski.

**Investigation:** Robert A. Johnson, Ronald L. Rutowski.

**Methodology:** Robert A. Johnson, Ronald L. Rutowski.

**Writing – original draft:** Robert A. Johnson.

**Writing – review & editing:** Ronald L. Rutowski.

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
