## [Decision Letter · Decision Letter 0]

7 Jan 2022

PONE-D-21-29154Color, activity period, and eye structure in four lineages of ants: pale, nocturnal species have evolved larger eyes and larger facets than their dark, diurnal congenersPLOS ONE

Dear Dr. Johnson,

Thank you for submitting your manuscript to PLOS ONE. After careful consideration, we feel that it has merit but does not fully meet PLOS ONE’s publication criteria as it currently stands. Therefore, we invite you to submit a revised version of the manuscript that addresses the points raised during the review process.

Both referees and I thought your study was highly relevant and tackle an important subject. However, the detailed review of the methods you used for the analysis of the data reveals that some points need clarifying, as well as the main hypotheses which do not seem to have been tested. In particular the link between period of activity and colour seems to be both a known fact that precedes the hypothesis and an hypothesis to test. The suggestion to use species instead of worker as statistical unit should be considered. Referee 2 makes a list of things to consider in the analyses and their interpretation which should be very carefully taken into account. This could mean that the main results and therefore the discussion may have to be rewritten accordingly (for example: the comment on line (561-562). I woul greatly appreciate a detailed response to the comments in the rebuttal letter.

We look forward to receiving your revised manuscript.

Kind regards,

Nicolas Chaline

Academic Editor

PLOS ONE

Journal Requirements:

3. Please amend your manuscript to include your abstract after the title page.

Reviewers' comments:

Reviewer's Responses to Questions

**Comments to the Author**

1. Is the manuscript technically sound, and do the data support the conclusions?

Reviewer #1: Partly

Reviewer #2: Partly

2. Has the statistical analysis been performed appropriately and rigorously? 

Reviewer #1: No

Reviewer #2: Yes

3. Have the authors made all data underlying the findings in their manuscript fully available?

Reviewer #1: Yes

Reviewer #2: Yes

4. Is the manuscript presented in an intelligible fashion and written in standard English?

Reviewer #1: Yes

Reviewer #2: Yes

5. Review Comments to the Author

Reviewer #1: I enjoyed the manuscript "Color, activity period, and eye structure in four lineages of ants: pale, nocturnal species have evolved larger eyes and larger facets than their dark, diurnal congeners." The text is straightforward and full of information about links between morphological traits and the species' natural history. The introduction is very well written. I would suggest relocating parts of the introduction to the methods to keep the manuscript in a more traditional format. However, as it did not interfere with understanding, and the text is so easy to follow, it is at the authors' choice.

My main concern was the use of workers as sampling units in the statistical analyses. Given that all hypotheses were postulated on the species' scale, the most correct would be to use the species as the replicate, not the workers. In the same line, I don't understand the connection between eye-related traits (not the ocelli) and worker size. I agreed that there is a general strong allometric relationship between size and eye shape, but the connection between the allometries and the hypothesis is not clear. The hypothesis deals with the variation of eye-related traits between species, not variation within species. Why not use the average of the traits divided by the average size of the species in a global analysis? In this approach, the species would be replicas, and the phylogenetic control (which I found highly relevant) could be included as a random variable (in a GLMM) or as a fixed factor (2-way GLM/ANOVA) like analysis. As the work investigated 23 species, the number of replicas would be sufficient to test the hypotheses.

My last comment is about the number of figures. Fourteen figures is a lot and end up drawing readers' attention from the most important findings. Some figures like figure 5 could be moved to the supplementary material, but I would try to move other figures as well (figure 14 is a strong candidate).

Small comments:

Lines 237-242: How did you establish which species were the closest?

Lines 409-411: When the interaction between continuous and categorical variable is significant, it is better to focus only on the interaction term.

Line 514: I would remove "and other", for clarity and parsimony.

Again, I commend the authors on an interesting manuscript.

Reviewer #2: Dear Dr. Johnson and PLOS ONE,

Thank you for the opportunity to contribute with the present manuscript. I want to highlight that I really appreciate the approach of the manuscript. The relationship between visual acuity and activity period in ants is an interesting topic that deserves more attention, and the link with cuticular brightness also revealed important aspects of the convergent evolution of associated traits under a common ecological aspect, in this case, nocturnal foranging.

I will highlight here just the main points that substantiate my decision. Minor comments are provided in the attached file of the manuscript.

INTRODUCTION

Line 62: I did not understand the reasoning of this hypothesis. It seems that the authors are assuming that pale ants are also nocturnal, so the authors are expecting that those visual parameters are larger in pale species. However, the relationship between intugment color and daily activity patterns is something that the authors also want to test in this work, so why are they providing such an assumption about visual parameters and intugment color before testing it? In theory the authors do not have elements to state it beforehand for the specific species they are addressing here, and this hypothesis should be related to the period of activity of the ant species being analyzed, and not their color.

METHODS

Lines 131-135: I have three main concerns regarding this metodology.

1. How did the authors handle the sparkles of light from the antweb images, which could occur exactly on the regions measured? A statement about this issues on the manuscript would be valuable, because anyone that take a look to the antweb images realize that the pictures have those sparkles of light that will influence the brightness value.

2. Did the authors tested if the brightness values differ between body regions of each species? Looking at figures 1-4 we see some species where one of the body regions is noticeable darker or more pale than the remaing regions, which would influence the brightness mean of each worker, and consequently the value for the species. An explanation about the reasoning to consider such a mean value from different body regions in the text would also be valuable.

3. Was it the same person that conducted all measurements? This could also be stated on the text.

Line 227: I have some issues about citing software guides as the main reference for any statistical analysis. I am not a user of SPSS, so I am not aware of how this specific guide exposes and discuss those analysis, but in general those guides are superficial and always recommend the consultation of more detailed statistical references for publication purposes. The intention of those guides is just to show how to perform the analysis in the software, nothing more. There is a bunch of exceptional statistical books available at any university, and also numerous reference papers, so I respectfully suggest the authors to consider such a reference for the statistical analysis.

Line 232: A reference about those assumptions would be valuable here.

Line 241: Why don't verifying the visual assessment of eye size?

RESULTS

Line 250: I suggest the authors to reconsider the presentation of this figure, it doesn't provide extra information in relation to Table 1, and those differences in brightness value can be clearly visualized in the table.

Line 287: I respectfully suggest a main rephrasing here, because statistical tests don't intend to be significant or not, the level of significance is predefined by who is conducting the test (following the recommendations of the scientific community) and is based on a predefined clearcut that doesn't influence the results of the test, only its interpretation. So, something like: "The result of the Levene's test suggests that the variances of the dependent variables were not homogeneous..."... will sound more technically correct.

This kind of statements appear several times along the explanation of results and even on the discussion section, and I did not highlight all of them on the attached file and will not do so here, but I strongly suggest the authors to review all of those sentences considering those arguments and the ones I will highlight in further comments.

Line 288: Why this P value? Is there a reference to substantiate this choice? Please clarify, because it is not clear why such a "correction" will solve the problem of non homogeneous variance.

Line 299: I respectfully suggest the authors to avoid statements like "varied significantly" and "were significantly higher...". Technically this don't make sense if you are considering a threshold value like P = 0.05 to reject or not the null hypothesis. If the statistical test suggests that there is no difference between treatments, you need to say that the variables don't vary, and not that there was no significant difference, because in fact there was no difference at all (based on your assumptions regarding the P value). Therefore, in case of rejection of the null hypothesis that there is no difference (in this case here, P < 0.05 or P < 0.01), you could just say that there was difference, one treatment was greater than the other, or something like that, and present the test results, without using "significantly". By the way, a look to this paper "Dushoff, J., Kain, M. P., & Bolker, B. M. (2019). I can see clearly now: reinterpreting statistical significance. Methods in Ecology and Evolution, 10(6), 756-759." can help to understand the meaning of significance and why the way we expose the results of statistical tests matters.

Line 315: Would the authors consider to joint this table with table 1? I understand that it could became a very huge table (add four more columns to table 1), but the information would be displayed in a more compact way, and it would be more easy to visualize the correlation between color, period of activity and eye measurements of the species.

Line 402: It is a little bit hard to get details about these results because the legends of the figures don't provide information about the line type for each species, but I have the impression that Veromessor lariversi does not increase D with mesossoma length, instead of V. smithi. Please check this issue, and sorry if I am wrong.

Line 408: The authors have a hypothesis regarding this parameter and the color/period of activity of the ant species. I suggest them to expose the results taking into account this hypothesis, like stating if the main results agree with what was expected.

Line 415 (but also many other instances along the Results section): I suggest the authors to provide the exact P values in all cases, it gives more transparency and is also a more standardized way to present P values than the way being provided along the manuscript (sometimes is P < .005, or P < 0.02, or P < 0.001, etc.).

Line 481: Respectfully, I think that this section doesn't make sense if the authors don't compare color and eye measurements between species classified as dark and pale. The authors measured brightness values for all those species, will it take too much time and effort to measure a general eye size measurement for those species and test if pale and dark coloured species vary in those variables according to the predictions? If this is the case, I think that this section should be removed from the paper. The way this section is presented, the reader should look by their own if the proposed hypothesis also agrees for these additional species. Finally, I cannot became convinced when "visual inspection of eye size" is the only information provided.

Lines 561-562: According to Table 1, there are other dark-colored species of Veromessor that are classified as crepuscular-nocturnal. So, how can the authors assure that V. smithi is the more nocturnal of the dark-colored species in the genus? Moreover, if the period of activity was so important for the evolution of eye size in the genus, how the authors explain that pale-colored species of the genus that are strictly nocturnal have smaller eyes than V. smithi? I suggest the authors to also consider alternative explanations to the large eye size of V. smithi along with its nocturnal activity, because there are exceptions on the genus, like V. stoddardi, which don't followed this pattern.

Lines 608-609: I disagree with this statement, no data was provided and no test was applied to assure that this statement is correct. If visual inspection of ant images is enough to state this correlation, why the authors measured eye parameters for the other species? Something like "suggest" instead of "revealed" sounds more realistic based on the methodology applied to substantiate this statement. However, consider the earlier comments about this section.

Lines 654-659: I failed to understand what is the functional contrast of ocelli between alates x workers. If in both the ocelli detects polarized light, there is no functional contrast, because flying x walking depends on other traits, not the ocelli.

Figures 6, 8, 9 and 10: Please provide information about the line type for each species on all figures.

6. PLOS authors have the option to publish the peer review history of their article (what does this mean?). If published, this will include your full peer review and any attached files.

Reviewer #1: No

Reviewer #2: No

---

## [Author Response · Author response to Decision Letter 0]

28 Mar 2022

5. Review Comments to the Author

First, we would like to thank both reviewers for their critical reading of the manuscript and constructive comments that we believe has increased the readability and understanding of the paper. We reply to each of their comments below in red, as well a reply to each comment on the pdf file that was included with the review. We have also made some minor editorial changes (which are in track changes) that we believe improve clarity and readaibility.

Reviewer #1: I enjoyed the manuscript "Color, activity period, and eye structure in four lineages of ants: pale, nocturnal species have evolved larger eyes and larger facets than their dark, diurnal congeners." The text is straightforward and full of information about links between morphological traits and the species' natural history. The introduction is very well written. I would suggest relocating parts of the introduction to the methods to keep the manuscript in a more traditional format. However, as it did not interfere with understanding, and the text is so easy to follow, it is at the authors' choice.

My main concern was the use of workers as sampling units in the statistical analyses. Given that all hypotheses were postulated on the species' scale, the most correct would be to use the species as the replicate, not the workers. In the same line, I don't understand the connection between eye-related traits (not the ocelli) and worker size. I agreed that there is a general strong allometric relationship between size and eye shape, but the connection between the allometries and the hypothesis is not clear. The hypothesis deals with the variation of eye-related traits between species, not variation within species. Why not use the average of the traits divided by the average size of the species in a global analysis? In this approach, the species would be replicas, and the phylogenetic control (which I found highly relevant) could be included as a random variable (in a GLMM) or as a fixed factor (2-way GLM/ANOVA) like analysis. As the work investigated 23 species, the number of replicas would be sufficient to test the hypotheses.

We appreciate that this reviewer thought of a possible alternative statistic to analyze our eye data in a more global fashion. However, we believe that our use of MANCOVA is the better analysis for this data. The primary reason for using MANCOVA is to account for and to include within species variation in the analyses, which allows for subsequent post-hoc tests of species differences for the three dependent eye variables. Thus, we can account for and test trait variation using data across the range of body sizes (the covariate) for each species, which is especially important given that several of these species have polymorphic workers. As mentioned by the reviewer, there often is a strong allometric relationship between size and eye shape, and MANCOVA accounts for and factors out this allometry when testing for species variation in these traits. Post-hoc differences across species within each genus can then be compared via all possible pairwise comparisons of species. Alternatively, using the ratio of (dependent variable/mesosoma length) (as suggested by the reviewer) provides only one data point per species, and consequently does not allow for further post-hoc testing of species variation in the dependent variables. Our understanding of the reviewer’s comments is that genus would be used as a fixed factor in a 2-way GLM ANOVA, which would show if there were differences across genera, but it does not allow for post-hoc comparisons across species within each genus.

Related to this comment, we have added information on our sampling regime for the included species. In the Body size and eye measurements subheading, we have added two sentences (in red) after the first sentence of the section (in black) to further explain our data collection. These sentences now read, “We measured body size and eye characteristics for workers from all 23 species listed above. These measurements included up to 15 workers per species, selected from up to seven colonies (when available), so as to include variation within and across colonies. Workers of several species are weakly to strongly polymorphic, and we selected workers within colonies of these species to span the variation in body size”. 

My last comment is about the number of figures. Fourteen figures is a lot and end up drawing readers' attention from the most important findings. Some figures like figure 5 could be moved to the supplementary material, but I would try to move other figures as well (figure 14 is a strong candidate).

We have deleted Figure 5. To provide a bit more data in the text, we have added ranges for both pale and dark species.

Small comments:

Lines 237-242: How did you establish which species were the closest?

This paragraph has been rewritten and expanded so as to include comparisons to dark species for verification of relative eye size. Please see response below for Reviewer #2 under Methods for query regarding line 241.

Lines 409-411: When the interaction between continuous and categorical variable is significant, it is better to focus only on the interaction term.

I believe that the reviewer misinterpreted this result, as both variables (genus and activity period) are categorical variables. We discuss both interaction and main effects. The first sentence of the paragraph (which discusses the continuous variable [ommatidial angle]) has been moved to the end of the paragraph to present a better topic sentence for the results of this subheading. 

Line 514: I would remove "and other", for clarity and parsimony.

As recommended in the pdf annotations by this reviewer, we have added potentially to this clause so that it now reads “suggesting that pale coloration is linked to nocturnal foraging in these and potentially other ants”.

Again, I commend the authors on an interesting manuscript.

Reviewer #2: Dear Dr. Johnson and PLOS ONE,

Thank you for the opportunity to contribute with the present manuscript. I want to highlight that I really appreciate the approach of the manuscript. The relationship between visual acuity and activity period in ants is an interesting topic that deserves more attention, and the link with cuticular brightness also revealed important aspects of the convergent evolution of associated traits under a common ecological aspect, in this case, nocturnal foranging.

I will highlight here just the main points that substantiate my decision. Minor comments are provided in the attached file of the manuscript.

INTRODUCTION

Line 62: I did not understand the reasoning of this hypothesis. It seems that the authors are assuming that pale ants are also nocturnal, so the authors are expecting that those visual parameters are larger in pale species. However, the relationship between integment color and daily activity patterns is something that the authors also want to test in this work, so why are they providing such an assumption about visual parameters and intugment color before testing it? In theory the authors do not have elements to state it beforehand for the specific species they are addressing here, and this hypothesis should be related to the period of activity of the ant species being analyzed, and not their color.

We have revised this paragraph to include two hypotheses regarding color, activity period, and eye structure – with the first hypothesis testing the association between body color and activity period, and the second the relationship between color/activity period and eye structure. The first two sentences have now been rewritten and expanded to read, “This study tests two hypotheses related to body color, activity period, and eye structure. As detailed above, depigmented and pale species tend to be restricted to lightless and dim light environments. Consequently, we first test the prediction that in four genera of ants activity period of pale species likewise is restricted to dim light conditions, i.e., nocturnal, whereas activity period of their dark congeners is not restricted to dim light conditions. To this end, we first quantified for each species the association between cuticular pigmentation and daily above-ground activity patterns, i.e., whether pale species are more likely to be nocturnal than their dark congeners. Second, we tested the prediction derived from the arguments above that eye structure varies with body color and activity time.”

METHODS

Lines 131-135: I have three main concerns regarding this methodology.

1. How did the authors handle the sparkles of light from the antweb images, which could occur exactly on the regions measured? A statement about this issues on the manuscript would be valuable, because anyone that take a look to the antweb images realize that the pictures have those sparkles of light that will influence the brightness value.

We have redone our brightness measures to account for various issues raised by the reviewer. First, we did not realize that there was a sample size selection for the brightness value in Photoshop. Consequently, we have redone and standardized all brightness measures using the 11 x 11 pixel function so that values are averaged over a larger area; this also reduces variation in B from the specific site that was measured. We also note the sparkle (reflectance) referred to by the reviewer, and our new B measurements avoided areas that displayed high reflectance. The text under the subheading “Measurement of body coloration” has been rewritten to address this comment and the two following comments on this methodology as follows and the text was also changed in the reviewer pdf file beginning on line 131. This paragraph now reads,

“All brightness, body size, and eye measurements were made from photographs of workers as described below. Brightness (B) was measured using the color window in Adobe Photoshop from photographs downloaded from Antweb (www.antweb.org). This assessment is similar to the lightness value in HSL that has been used to characterize body color in other studies of ants [1, 2]. We excluded specimens that were obviously discolored, i.e., those in which the color differed substantially from intraspecific specimens recently collected by RAJ. Using the photograph of the worker body in profile, we measured the average B over an 11 × 11 pixel area (ca. 8.2 mm2) on the head (immediately posterior to the eye), mesosoma (center of mesopleura), and gaster (anteroposterior portion of first gastral tergum), then averaged these values for each worker, then averaged that value across all workers for each species. In measuring B, we focused on areas of diffuse reflectance and avoided areas of specular (mirror-like) reflectance so as to minimize their effect on B values. Using the average B value across the three body tagmata accounted for intra-individual variation in color that occurs in many ant species. All measurements were made by RAJ. We compared using a t-test mean B values for species qualitatively categorized as pale versus dark.”

Remeasuring of B changed those values for all species, but all values remained within a narrow window that was similar to our previous measurements. Additionally, the overall results for comparisons between pale and dark species remained the same.

In line with this comment, we have updated the mean B values and added ranges for values of pale versus dark species (lines 249-250 in pdf).

2. Did the authors tested if the brightness values differ between body regions of each species? Looking at figures 1-4 we see some species where one of the body regions is noticeable darker or more pale than the remaing regions, which would influence the brightness mean of each worker, and consequently the value for the species. An explanation about the reasoning to consider such a mean value from different body regions in the text would also be valuable.

This comment is addressed in above reply.

3. Was it the same person that conducted all measurements? This could also be stated on the text.

Yes, the same person conducted all measurement (RAJ), and this has been added to the text (see above).

Line 227: I have some issues about citing software guides as the main reference for any statistical analysis. I am not a user of SPSS, so I am not aware of how this specific guide exposes and discuss those analysis, but in general those guides are superficial and always recommend the consultation of more detailed statistical references for publication purposes. The intention of those guides is just to show how to perform the analysis in the software, nothing more. There is a bunch of exceptional statistical books available at any university, and also numerous reference papers, so I respectfully suggest the authors to consider such a reference for the statistical analysis. 

We have added the following two references that refer to MANCOVA and its assumptions.

1. Bishop TR, Robertson MP, Gibb H, van Rensburg BJ, Braschler B, Chown SL, et al. Ant assemblages have darker and larger members in cold environments. Global Ecol Biogeogr. 2016;25:1489–99.

2. Law SJ, Bishop TR, Eggleton P, Griggiths H, Ashton L, Parr C. Darker ants dominate the canopy: testing macroecological hypotheses for patterns in colour along a microclimatic gradient. J Anim Ecol. 2020;89:348–59.

3. Greiner B. Adaptations for nocturnal vision in insect apposition eyes. Int Rev Cytol. 2006;250:1–46.

4. Stöckl A, Smolka J, O'Carroll D, Warrant E. Resolving the trade-off between visual sensitivity and spatial acuity - lessons from hawkmoths. Integrat Comp Biol. 2017;57:1093–103.

5. Laerd Statistics. One-way MANCOVA in SPSS statistics: Lund Research Ltd,; 2018 [cited 2022 February 18]. Available from: https://statistics.laerd.com/spss-tutorials/one-way-mancova-using-spss-statistics.php.

Line 232: A reference about those assumptions would be valuable here. 

See reply from above comment

Line 241: Why don't verifying the visual assessment of eye size?

This section of the methods has been rewritten and expanded for clarification, and also to include verification of relative eye size for pale species. This paragraph now reads, “We used Antweb (www.antweb.org) to scan photographs for pale ant species in the genera Aphaenogaster, Crematogaster, Messor, and Temnothorax (subfamily Myrmicinae), and Dorymyrmex and Iridomyrmex (subfamily Dolichoderinae). We scrolled through frontal photographs of the head for all species in each genus looking for species that appeared pale. We verified our visual assessment of color for these taxa by measuring their brightness value (B) using Adobe Photoshop, as detailed above, with pale species being those with a B value > 70. We then assessed relative eye size for these species, calculated as eye area/mesosoma length, compared to their dark congeners that all had B values < 70. Phylogenies were not available for these genera, so we used the first five species/nominal subspecies on the Antweb page from each genus. Pale species with enlarged eyes were those that had a relative eye size larger than that of all five dark congeners. Photographs of each species can be viewed by going to https://www.antweb.org/advSearch.do, then placing the genus and species name in the advanced search box.” 

We also note that we have revamped Table 5 after measuring B values and relative eye size on pale species. We have deleted the CASENT persistent identifier column and replaced it with “Relative eye size”, which we use as our measure of enlarged eyes. In conjunction with this, we have added a supplementary Table (S1 Table), which have B values and relative eye size for dark congeners in these genera.

RESULTS

Line 250: I suggest the authors to reconsider the presentation of this figure, it doesn't provide extra information in relation to Table 1, and those differences in brightness value can be clearly visualized in the table.

Figure 5 has been deleted from the manuscript, and added as a supplementary Figure – S1 Figure.

Line 287: I respectfully suggest a main rephrasing here, because statistical tests don't intend to be significant or not, the level of significance is predefined by who is conducting the test (following the recommendations of the scientific community) and is based on a predefined clearcut that doesn't influence the results of the test, only its interpretation. So, something like: "The result of the Levene's test suggests that the variances of the dependent variables were not homogeneous..."... will sound more technically correct.

This kind of statements appear several times along the explanation of results and even on the discussion section, and I did not highlight all of them on the attached file and will not do so here, but I strongly suggest the authors to review all of those sentences considering those arguments and the ones I will highlight in further comments.

We have changed the wording of statistical tests to reflect the comment. As suggested, we have used differed instead of significant and well as editing other similar phrasing.

Line 288: Why this P value? Is there a reference to substantiate this choice? Please clarify, because it is not clear why such a "correction" will solve the problem of non homogeneous variance.

This problem has been resolved by finding a better transformation that better satisfies the problem with Levene’s test. We now use P < 0.05 for this post-hoc test. 

Line 299: I respectfully suggest the authors to avoid statements like "varied significantly" and "were significantly higher...". Technically this don't make sense if you are considering a threshold value like P = 0.05 to reject or not the null hypothesis. If the statistical test suggests that there is no difference between treatments, you need to say that the variables don't vary, and not that there was no significant difference, because in fact there was no difference at all (based on your assumptions regarding the P value). Therefore, in case of rejection of the null hypothesis that there is no difference (in this case here, P < 0.05 or P < 0.01), you could just say that there was difference, one treatment was greater than the other, or something like that, and present the test results, without using "significantly". By the way, a look to this paper "Dushoff, J., Kain, M. P., & Bolker, B. M. (2019). I can see clearly now: reinterpreting statistical significance. Methods in Ecology and Evolution, 10(6), 756-759." can help to understand the meaning of significance and why the way we expose the results of statistical tests matters.

We have rewritten this section in accord with the reviewer comments. We have largely removed the word “significant” and replaced it with “differed”, except when presenting covariate results. We have also changed to exact P values throughout except for very low P values that are all presented as P < 0.001.

Line 315: Would the authors consider to joint this table with table 1? I understand that it could became a very huge table (add four more columns to table 1), but the information would be displayed in a more compact way, and it would be more easy to visualize the correlation between color, period of activity and eye measurements of the species.

We have joined Tables 1 and 4 into one large Table (Table 1), as suggested by the reviewer.

Line 402: It is a little bit hard to get details about these results because the legends of the figures don't provide information about the line type for each species, but I have the impression that Veromessor lariversi does not increase D with mesossoma length, instead of V. smithi. Please check this issue, and sorry if I am wrong.

We have added line type information to the figure caption of all relevant figures

Line 408: The authors have a hypothesis regarding this parameter and the color/period of activity of the ant species. I suggest them to expose the results taking into account this hypothesis, like stating if the main results agree with what was expected.

We have changed the text accordingly relative to our hypotheses, but we have made these changes in the Discussion section rather than the results. At the first line of the discussion we have added, “as predicted.” Additionally, the first paragraph of the next subheading (Compound eye morphology) has been rewritten. The last sentence was deleted and moved in part to the topic sentence. The first two sentences of this paragraph now read, “We hypothesized that the eyes of pale ant species maximized sensitivity at the expense of resolution, which is the pattern typical for most insects with apposition eyes [3, 4]. These features to enhance sensitivity include a larger eye size, larger facets, larger Δϕ‘s, and a higher �. “ Following that we detail the results relative to the hypotheses.

Line 415 (but also many other instances along the Results section): I suggest the authors to provide the exact P values in all cases, it gives more transparency and is also a more standardized way to present P values than the way being provided along the manuscript (sometimes is P < .005, or P < 0.02, or P < 0.001, etc.).

We have changed the manuscript by including exact P values throughout except for very low P values that are all presented as P < 0.001.

Line 481: Respectfully, I think that this section doesn't make sense if the authors don't compare color and eye measurements between species classified as dark and pale. The authors measured brightness values for all those species, will it take too much time and effort to measure a general eye size measurement for those species and test if pale and dark coloured species vary in those variables according to the predictions? If this is the case, I think that this section should be removed from the paper. The way this section is presented, the reader should look by their own if the proposed hypothesis also agrees for these additional species. Finally, I cannot became convinced when "visual inspection of eye size" is the only information provided.

We have modified this section as per the reviewer comment. See the reply given above regarding line 241.

Lines 561-562: According to Table 1, there are other dark-colored species of Veromessor that are classified as crepuscular-nocturnal. So, how can the authors assure that V. smithi is the more nocturnal of the dark-colored species in the genus? Moreover, if the period of activity was so important for the evolution of eye size in the genus, how the authors explain that pale-colored species of the genus that are strictly nocturnal have smaller eyes than V. smithi? I suggest the authors to also consider alternative explanations to the large eye size of V. smithi along with its nocturnal activity, because there are exceptions on the genus, like V. stoddardi, which don't followed this pattern.

This sentence has been changed to, "This may result from the fact that V. smithi is a nocturnally-active dark species." A possible explanation for this variation is given later in this same paragraph, as well as mentioning these particular species in regard to variation within Veromessor.

Lines 608-609: I disagree with this statement, no data was provided and no test was applied to assure that this statement is correct. If visual inspection of ant images is enough to state this correlation, why the authors measured eye parameters for the other species? Something like "suggest" instead of "revealed" sounds more realistic based on the methodology applied to substantiate this statement. However, consider the earlier comments about this section.

We have now included comparative data on brightness and eye size for pale species that we found on Antweb that we used to compare those pale species with their dark congeners. We have also rewritten this topic sentence to say, “Our survey of images on Antweb found numerous additional species of pale ants with enlarged eyes in the clades studied here and in other genera compared to their dark congeners.”

Lines 654-659: I failed to understand what is the functional contrast of ocelli between alates x workers. If in both the ocelli detects polarized light, there is no functional contrast, because flying x walking depends on other traits, not the ocelli.

This paragraph has been revised to increase its clarity. It now reads, "The function of ocelli in ant workers is poorly understood because most species lack ocelli (notable exceptions include the genera Cataglyphis, Formica, Myrmecocystus, Polyergus in the subfamily Formicinae; Myrmecia in the subfamily Myrmeciinae; Pseudomyrmex in the subfamily Pseudomyrmecinae). Flying insects have three ocelli that serve the general purpose of sensing polarized light for navigation and maintaining flight stability, whereas worker ants that have been studied use their ocelli to detect polarized light for navigation in Cataglyphis bicolor [87] or to gather light in Myrmecia [86]. Ocelli almost always are present in the flying queen and male ants, such that species in which the pedestrian workers possess ocelli provide an avenue to compare ocellar structure and function relative to mode of locomotion and caste. The only such comparison using the nocturnal Myrmecia nigriceps found structural differences between workers and males that included presence of a tapetum in males but not in workers [86]".

Figures 6, 8, 9 and 10: Please provide information about the line type for each species on all figures.

Line types have been added to the plots within each of these figures

6. PLOS authors have the option to publish the peer review history of their article (what does this mean?). If published, this will include your full peer review and any attached files.

Do you want your identity to be public for this peer review? For information about this choice, including consent withdrawal, please see our Privacy Policy.

Reviewer #1: No

Reviewer #2: No

[5]

---

## [Decision Letter · Decision Letter 1]

24 May 2022

PONE-D-21-29154R1Color, activity period, and eye structure in four lineages of ants: pale, nocturnal species have evolved larger eyes and larger facets than their dark, diurnal congenersPLOS ONE

Dear Dr. Johnson,

Thank you for submitting your manuscript to PLOS ONE. After careful consideration, we feel that it has merit but does not fully meet PLOS ONE’s publication criteria as it currently stands. Therefore, we invite you to submit a revised version of the manuscript that addresses the points raised during the review process.

Your MS is very much improved adn almost ready for publication, congratuilations. However, I would ask you to conider the minor changes suggested taht would make your MS better stillPlease ensure that your decision is justified on PLOS ONE’s publication criteria and not, for example, on novelty or perceived impact.

We look forward to receiving your revised manuscript.

Kind regards,

Nicolas Chaline

Academic Editor

PLOS ONE

Journal Requirements:

Reviewers' comments:

Reviewer's Responses to Questions

**Comments to the Author**

1. If the authors have adequately addressed your comments raised in a previous round of review and you feel that this manuscript is now acceptable for publication, you may indicate that here to bypass the “Comments to the Author” section, enter your conflict of interest statement in the “Confidential to Editor” section, and submit your "Accept" recommendation.

Reviewer #1: All comments have been addressed

Reviewer #2: (No Response)

2. Is the manuscript technically sound, and do the data support the conclusions?

Reviewer #1: (No Response)

Reviewer #2: Yes

3. Has the statistical analysis been performed appropriately and rigorously? 

Reviewer #1: (No Response)

Reviewer #2: I Don't Know

4. Have the authors made all data underlying the findings in their manuscript fully available?

Reviewer #1: (No Response)

Reviewer #2: Yes

5. Is the manuscript presented in an intelligible fashion and written in standard English?

Reviewer #1: (No Response)

Reviewer #2: Yes

6. Review Comments to the Author

Reviewer #1: Dear authors,

Thanks for the clarification and changes to the main text. For some reason, I missed the interest in intraspecific variation, which is now much clearer. I had already liked the manuscript in the first round, and I believe this new version is even more precise. Again congratulations on the interesting work.

Reviewer #2: I congratulate the authors by the improved version of the manuscript provided. Most of my first appointments were satisfactorily met, and I just have minor recommendations for this new version, as follows:

Lines 77-79: Given that the authors also didn’t control explicitly for the phylogenetic relationships of the species in their statistical analysis, I would say that they cannot advocate that here they are avoiding completely those issues, as this sentence could suggest. I would suggest a little rephrasing here to avoid the idea that in the present manuscript phylogenetic corrections were explicitly performed.

Line 93: In “how closely the eye is constructed” I would replace “is constructed” by “develops”.

Lines 113-114: This predictive hypothesis need to be reviewed, since the authors did not test the relationship between brightness and activity period for this additional species.

Lines 125-128: for the remaining genera considered, the author justify the choice of the species by their phylogenetic relationships, but here for Myrmecocystus the choice was based on body size, and the phylogenetic relationships among the choose pairs of species seems not to follow a clear pattern as in the other genera. What was the criteria to choose those species pairs?

Line 329: Don't seems to be the case for many of them (P < 0.05), specially in the case of Box’s M test, but there are also problems with some Levene's tests and Wilk's lambda. I suggest the authors to review this statement and the tests to be sure if the assumptions were really met.

Lines 366-368: Here again it is not clear what is the meaning of the significant ANCOVA. The manuscript is large and refers to many different results and tests, so if the reader should return to the methods section every time a statistical test is present in the results section to check what exactly the test means, the reading would be exhausting. I suggest the authors to present the results in a more straightforward way, indicating exactly what the result of the test means (like “the test suggest that individuals differ in size (results of the test)” or “the test indicates that the diameter of the ocellus is greater in pale than dark species (results of the test)”. Although the authors already improved those issues that I pointed in the first review, which I comply for the effort, it could still be improved in some instances as this one.

Lines 446-451: The same as the previous comment. I would suggest to rewrite this sentence as “Genus has an important effect on Δϕ (F3,32 = 10.1, P < 0.001). Although the isolated effect of activity period over Δϕ showed a weak effect (F1,32 = 4.0, P = 0.055), the interaction of genus × activity period showed a strong effect (F3,32 = 7.3, P < 0.001), which indicates that the four genera differed in the direction and magnitude of differences in Δϕ”.

Lines 456-459: Another rewriting suggestion to inform the results in a more straightforward manner and reduce text length: “Mesosoma length was an important covariate in the above model (F1,31 = 5.4, P = 0.026), a fact largely influenced by Δϕ decreasing in larger workers of Temnothorax and Veromessor, and not varying with mesosoma length within species of Myrmecocystus and Aphaenogaster (Fig 10).”.

To reinforce, those suggestions to rewrite statements describing the results should be taken only as suggestions, the authors can decide to not follow them or to apply them throughout the results section if they agree that it would improve the readability of the manuscript.

Lines 466-468: There is no strong effect (difference in p based on P-value) between pale and dark species in those genera except in Myrmecocystus, I suggest the authors to take a look on it.

Lines 545-546: The relationship between body color and activity period does not support the hypothesis that activity pattern produces strong selection patterns in eye structure, please rewrite this statement.

Line 599: V. smithi should be in bold face, right?

Line 614: The symbol for micrometers is not correct, please take a look on it.

Lines 621-623: It is not clear here which genera the authors are referring by “latter three genera”.

Line 676: There are images of M. pyramicus available in AntWeb.

Tables

Table 1: I suggest the authors to indicate in the table that “facet diameter” is the variable D, since this is how they refer to this variable throughout most of the text.

Table 4: Please replace Mauchley’s by Mauchly’s.

7. PLOS authors have the option to publish the peer review history of their article (what does this mean?). If published, this will include your full peer review and any attached files.

Reviewer #1: No

Reviewer #2: No

---

## [Author Response · Author response to Decision Letter 1]

14 Jul 2022

Reviewers' comments:

Reviewer's Responses to Questions

Comments to the Author

1. If the authors have adequately addressed your comments raised in a previous round of review and you feel that this manuscript is now acceptable for publication, you may indicate that here to bypass the “Comments to the Author” section, enter your conflict of interest statement in the “Confidential to Editor” section, and submit your "Accept" recommendation.

Reviewer #1: All comments have been addressed

Reviewer #2: (No Response)

2. Is the manuscript technically sound, and do the data support the conclusions?

Reviewer #1: (No Response)

Reviewer #2: Yes

3. Has the statistical analysis been performed appropriately and rigorously? 

Reviewer #1: (No Response)

Reviewer #2: I Don't Know

4. Have the authors made all data underlying the findings in their manuscript fully available?

Reviewer #1: (No Response)

Reviewer #2: Yes

5. Is the manuscript presented in an intelligible fashion and written in standard English?

Reviewer #1: (No Response)

Reviewer #2: Yes

6. Review Comments to the Author

Reviewer #1: Dear authors,

Thanks for the clarification and changes to the main text. For some reason, I missed the interest in intraspecific variation, which is now much clearer. I had already liked the manuscript in the first round, and I believe this new version is even more precise. Again congratulations on the interesting work.

Reviewer #2: I congratulate the authors by the improved version of the manuscript provided. Most of my first appointments were satisfactorily met, and I just have minor recommendations for this new version, as follows:

Lines 77-79: Given that the authors also didn’t control explicitly for the phylogenetic relationships of the species in their statistical analysis, I would say that they cannot advocate that here they are avoiding completely those issues, as this sentence could suggest. I would suggest a little rephrasing here to avoid the idea that in the present manuscript phylogenetic corrections were explicitly performed.

We have added text to the end of this paragraph to indicate that we did not perform phylogenetic corrections. The added text reads, “In this study, phylogenies were available for all four examined ant genera, such that we could compare eye structure of closely related pale and dark species. However, phylogenetic corrections were not performed”.

Line 93: In “how closely the eye is constructed” I would replace “is constructed” by “develops”.

Changed as suggested

Lines 113-114: This predictive hypothesis need to be reviewed, since the authors did not test the relationship between brightness and activity period for this additional species.

We have revised the last sentence of the paragraph as per the suggestion of the reviewer. The sentence previously read, “Here again, we expected pale body color to be correlated with nocturnal activity as well as eye morphology that enhances visual sensitivity”. 

The sentence has been revised to read, “We could not test the correlation between body color and nocturnal activity in these species, but we expect pale body color to be correlated with eye morphology that enhances visual sensitivity”.

Lines 125-128: for the remaining genera considered, the author justify the choice of the species by their phylogenetic relationships, but here for Myrmecocystus the choice was based on body size, and the phylogenetic relationships among the choose pairs of species seems not to follow a clear pattern as in the other genera. What was the criteria to choose those species pairs?

The first paragraph discussing Myrmecocystus and species selection has been rewritten for clarity and additional information on species selection. The paragraph now reads, “Myrmecocystus: We examined 74 workers from six species (Fig 1). This genus is restricted to North America, and it consists of 29 described species (Snelling 1982, 1976), plus several undescribed and cryptic species (van Elst et al. 2021). Species of Myrmecocystus display a wide range of body sizes, and all species are size polymorphic. We compared three size-similar species pairs that differed in pigmentation across this range of body sizes: small (M. christineae and M. yuma), medium (M. navajo and M. kennedyi), and large (M. mexicanus-02 and M. mendax-03). The genus consists of two major clades: most pale species occur in one clade, and the rest of the pale species and all dark species occur in the other clade. The latter clade consists of six smaller clades – one includes the rest of the pale species, and the other five consist of dark species (van Elst et al. 2021). Consequently, we could not compare eye structure for phylogenetically adjacent pale and dark species. Instead, we chose both pale and dark species pairs based on a combination of availability of specimens, taxonomic stability, and relative ease of identification”. 

Line 329: Don't seems to be the case for many of them (P < 0.05), specially in the case of Box’s M test, but there are also problems with some Levene's tests and Wilk's lambda. I suggest the authors to review this statement and the tests to be sure if the assumptions were really met.

We have reanalyzed the data for Temnothorax to account for the statistics not meeting the model assumptions for the homogeneity of regression slopes. Data were transformed again to help meet the assumptions, but problems still existed. Consequently, we redid the analysis in a slightly different way such that the assumptions were met. Additionally, note that we use an alpha value of P < 0.001 to denote significance for the Box’s M test, which is in accord with discussions of this statistic – references are added in the text to support this assertion. We also added a discussion regarding Veromessor and model assumptions. Text regarding the analysis for Veromessor is presented following the text for Temnothorax.

We have changed the text under the subheading “Eye area, facet number, and facet diameter” to describe our statistical methods for Temnothorax. The sentence, “Dependent variables met both assumptions for all four genera (Table 3)”, has been replaced by the following, “Box’s M test for homogeneity of variance-covariance matrices is not robust to violations of this assumption such that an alpha level of P < 0.001 is recommended (Hahs-Vaughn 2017; Tabachnick & Fidell 2001). Myrmecocystus and Aphaenogaster met both assumptions. Temnothorax met the homogeneity of variance-covariance assumption but not the assumption for homogeneity of regression slopes because the slopes for eye area differed across species (P = 0.007) (Table 3). Consequently, we reran the model after deleting eye area, and the homogeneity of regression slopes assumption was met using the two other dependent variables (facet number, facet diameter) (Table 3). We then analyzed variation in eye area across species separately using a one-way ANOVA. The dependent variable in this model was relative eye size, calculated as eye area/mesosoma length, which standardized eye area for body size (see above). Veromessor did not meet the homogeneity of variance-covariance assumption, so we used Pillai’s trace instead of Wilks’lambda to evaluate multivariate significance (Olson 1797). The Levene’s test for Veromessor was also significant for the regression slopes comparison and in the final model. Consequently, we decreased the P value for post-hoc tests to P < 0.01 (Tabachnick & Fidell 2001).

The following references regarding these statistical methods were added to the text.

Hahs-Vaughn DL. 2017. Applied multivariate statistical concepts. Routledge, New York.

Olson CL. 1797. Practical considerations in choosing a MANOVA test statistic: a rejoinder to Stevens. Psychological Bulletin 86, 1350–1352.

Tabachnick BG, Fidell LS. Using multivariate statistics. 6th ed. Upper Saddle River, New 

 Jersey: Pearson; 2014.

Accordingly, we have also added information regarding Temnothorax to the caption of Table 3 and to Table 3 itself. To the end of the caption we have added, “For Temnothorax, the first line is for all three variables, the second line is the model after deleting eye area (see text); values in the second line for the Levene’s test are for facet number and facet diameter, respectively”. We also note in this Table legend that Pillai’s trace was used instead of Wilks’ lambda for the multivariate test for Veromessor, which was also noted in the above revised methods.

The results are basically the same as before, but the section on Temnothorax has been revised to reflect the different analysis that is now used. The section for Temnothorax now reads, “Eye structure (facet number, facet diameter) differed among species of Temnothorax after controlling for mesosoma length (MANCOVA: Wilks’ λ= 0.054, F4,48 = 39.5, P < 0.001). The tests of between-subject effects demonstrated that both facet number and facet diameter differed across species (facet number: F2,25 = 20.8, P < 0.001; mean D: F2,25 = 53.4, P < 0.001; Fig 8). Based on estimated marginal means, pairwise comparisons across all species pairs using a LSD test showed that mean D was larger for the pale T. sp. BCA-5 than for the two dark congeners, and that facet number was higher in T. tricarinatus than for both T. neomexicanus and T. sp. BCA-5 (Fig 8). Relative eye area also differed across species (one-way ANOVA: F2,26 = 44.0, P < 0.001), with size being largest for T. sp. BCA-05, intermediate for T. tricarninatus, and smallest for T. neomexicanus (LSD test, P < 0.05; Fig 8). Mean D (using estimated marginal means) was 1.23× larger for T. sp. BCA-5 (17.76 μm) compared to T. neomexicanus (14.43 μm) and 1.32× larger than that for T. tricarinatus (13.42 μm) (Table 1).

Mesosoma length was an important covariate in the model (Wilks’ λ = 0.546, F2,24 = 9.97, P < 0.001), and tests of between-subjects effects differed for facet number (F1,25 = 20.8, P < 0.001), but not for mean D (F1,25 = 3.9, P = 0.058). Eye area and facet number increased with body size within all three species, while mean D increased with body size for T. sp. BCA-5 and T. tricarinatus, but it decreased with body size for T. neomexicanus (Fig 8).

In regard to Veromessor, the results and graphs have been rewritten accordingly using an alpha value of P < 0.01 

Lines 366-368: Here again it is not clear what is the meaning of the significant ANCOVA. The manuscript is large and refers to many different results and tests, so if the reader should return to the methods section every time a statistical test is present in the results section to check what exactly the test means, the reading would be exhausting. I suggest the authors to present the results in a more straightforward way, indicating exactly what the result of the test means (like “the test suggest that individuals differ in size (results of the test)” or “the test indicates that the diameter of the ocellus is greater in pale than dark species (results of the test)”. Although the authors already improved those issues that I pointed in the first review, which I comply for the effort, it could still be improved in some instances as this one.

The first sentence of this paragraph has been rewritten for additional clarity and now reads, “Diameter of the anterior ocellus differed across species after controlling for mesosoma length (ANCOVA: tests of between-subject effects; F7,98 = 69.6, P < 0.001; Fig. 6), but the species × mesosoma length interaction term was not significant (F7,91 = 0.96, P = 0.47)”. 

Lines 446-451: The same as the previous comment. I would suggest to rewrite this sentence as “Genus has an important effect on Δϕ (F3,32 = 10.1, P < 0.001). Although the isolated effect of activity period over Δϕ showed a weak effect (F1,32 = 4.0, P = 0.055), the interaction of genus × activity period showed a strong effect (F3,32 = 7.3, P < 0.001), which indicates that the four genera differed in the direction and magnitude of differences in Δϕ”.

This sentence has been rewritten as suggested by the reviewer. The paragraph now reads, “Genus has an important effect on Δϕ (ANCOVA: F3,32 = 10.1, P < 0.001). Although the isolated effect of activity period over Δϕ showed a weak effect (F1,32 = 4.0, P = 0.055), the interaction of genus × activity period showed a strong effect (F3,32 = 7.3, P < 0.001), which indicates that the four genera differed in the direction and magnitude of differences in Δϕ. Across all species, values of Δϕ ranged from 3.5–7° (Fig 10). 

Lines 456-459: Another rewriting suggestion to inform the results in a more straightforward manner and reduce text length: “Mesosoma length was an important covariate in the above model (F1,31 = 5.4, P = 0.026), a fact largely influenced by Δϕ decreasing in larger workers of Temnothorax and Veromessor, and not varying with mesosoma length within species of Myrmecocystus and Aphaenogaster (Fig 10).”.

To reinforce, those suggestions to rewrite statements describing the results should be taken only as suggestions, the authors can decide to not follow them or to apply them throughout the results section if they agree that it would improve the readability of the manuscript.

I also believe that the comment/edit by the reviewer makes the manuscript clearer. We have made the change as suggested to this comment. We have also edited other paragraphs that discuss mesosoma length as a covariate so as to have similar construction.

Lines 466-468: There is no strong effect (difference in p based on P-value) between pale and dark species in those genera except in Myrmecocystus, I suggest the authors to take a look on it.

We have edited the second part of this paragraph as suggested. It now reads: “Pale species had the larger mean � in Myrmecocystus (t-test: t8 df = -8.9, P < 0.001), while values did not differ between pale and dark species in the other three genera (Veromessor: t8 = -0.3, P = 0.74; Temnothorax: t4 = -1.7, P = 0.16; Aphaenogaster: t8 = -2.2, P = 0.058). Across genera � was highest for Aphaenogaster, intermediate for Temnothorax and Veromessor, and lowest for Myrmecocystus (Tukey’s HSD test, P < 0.05; Fig 11)”.

Lines 545-546: The relationship between body color and activity period does not support the hypothesis that activity pattern produces strong selection patterns in eye structure, please rewrite this statement.

We are not sure where the reviewer is going with this? We believe that activity period is the precursor to color and eye structure, such that activity patterns is the first trait to drive these other changes. We have rewritten this sentence slightly so that it now reads, “The above two patterns support the hypothesis that activity pattern can produce strong selection pressure on body coloration and eye structure”.

Line 599: V. smithi should be in bold face, right?

Yes, it should be bold font, and it has been changed.

Line 614: The symbol for micrometers is not correct, please take a look on it.

This has been changed to μm

Lines 621-623: It is not clear here which genera the authors are referring by “latter three genera”.

This sentence has been changed for better clarity. It previously read, “The lack of consistent patterns across the latter three genera likely …”. It now reads, “The lack of consistent patterns across species of Aphaenogaster, Temnothorax and Veromessor likely …”

Line 676: There are images of M. pyramicus available in AntWeb.

Yes, we are aware that there images of M. pyramicus on ANTWEB. For all other species that we included, we had two or more long series of workers to examine for ocelli and variation in presence/absence across all sizes, etc. ANTWEB only has photos of individual workers of M. pyramicus and we thus preferred to omit this species because of the limited sample size and number of colonies to examine.

Tables

Table 1: I suggest the authors to indicate in the table that “facet diameter” is the variable D, since this is how they refer to this variable throughout most of the text.

D has been added to subheading in Table 1

Table 4: Please replace Mauchley’s by Mauchly’s. 

Mauchley’s changed to Mauchly’s

7. PLOS authors have the option to publish the peer review history of their article (what does this mean?). If published, this will include your full peer review and any attached files.

Do you want your identity to be public for this peer review? For information about this choice, including consent withdrawal, please see our Privacy Policy.

Reviewer #1: No

Reviewer #2: No

---

## [Decision Letter · Decision Letter 2]

25 Aug 2022

Color, activity period, and eye structure in four lineages of ants: pale, nocturnal species have evolved larger eyes and larger facets than their dark, diurnal congeners

PONE-D-21-29154R2

Dear Dr. Johnson,

We’re pleased to inform you that your manuscript has been judged scientifically suitable for publication and will be formally accepted for publication once it meets all outstanding technical requirements.

Kind regards,

Nicolas Chaline

Academic Editor

PLOS ONE

Additional Editor Comments (optional):

Reviewers' comments:

Reviewer's Responses to Questions

**Comments to the Author**

1. If the authors have adequately addressed your comments raised in a previous round of review and you feel that this manuscript is now acceptable for publication, you may indicate that here to bypass the “Comments to the Author” section, enter your conflict of interest statement in the “Confidential to Editor” section, and submit your "Accept" recommendation.

Reviewer #1: All comments have been addressed

Reviewer #2: All comments have been addressed

2. Is the manuscript technically sound, and do the data support the conclusions?

Reviewer #1: (No Response)

Reviewer #2: Yes

3. Has the statistical analysis been performed appropriately and rigorously? 

Reviewer #1: (No Response)

Reviewer #2: Yes

4. Have the authors made all data underlying the findings in their manuscript fully available?

Reviewer #1: (No Response)

Reviewer #2: Yes

5. Is the manuscript presented in an intelligible fashion and written in standard English?

Reviewer #1: (No Response)

Reviewer #2: Yes

6. Review Comments to the Author

Reviewer #1: I had already recommended acceptance of the manuscript in the second round of review, but I agree that the points raised by reviewer #2 were essential to make the text clearer and more fluid. I congratulate the authors by the interesting work.

Reviewer #2: I congratulate the authors for their effort to provide this improved version of the manuscript. It is an interesting manuscript that will enhance substantially our knowledge about the evolution of eye characteristics in ant workers and how they relate to the ecology of species.

7. PLOS authors have the option to publish the peer review history of their article (what does this mean?). If published, this will include your full peer review and any attached files.

Reviewer #1: No

Reviewer #2: No

---

## [Editor Report · Acceptance letter]

13 Sep 2022

PONE-D-21-29154R2 

Color, activity period, and eye structure in four lineages of ants: pale, nocturnal species have evolved larger eyes and larger facets than their dark, diurnal congeners 

Dear Dr. Johnson:

I'm pleased to inform you that your manuscript has been deemed suitable for publication in PLOS ONE. Congratulations! Your manuscript is now with our production department. 

Kind regards, 

on behalf of

Professor Nicolas Chaline 

Academic Editor

PLOS ONE